| Open Peer Review | Microbial Genetics | Methods and Protocols

# Development of a culture-independent whole-genome sequencing of Nipah virus using the MinION Oxford Nanopore platform

Md. Mahfuzur Rahman,[1,2] Mojnu Miah,[1] Mohammad Enayet Hossain,[1] Samiur Rahim,[1] Sharmin Sultana,[3] Syed Moinuddin Satter,[1] Ariful Islam,[4] Shannon L. M. Whitmer,[5] Jonathan H. Epstein,[4] Christina F. Spiropoulou,[5] John D. Klena,[5] Tahmina Shirin,[3] Joel M. Montgomery,[5] Maria E. Kaczmarek,[4] Mohammed Ziaur Rahman,[1] Iqbal Kabir Jahid[2]

**ABSTRACT** Nipah virus (NiV) is a deadly zoonotic pathogen in Southeast Asia causing severe respiratory and encephalitis symptoms with a high fatality rate. Whole-genome sequencing (WGS) is crucial for tracking transmission, conducting epidemiological analyses, and understanding NiV's adaptive evolution. WGS is essential for analyzing genomes, particularly in understanding pathogen nature, and pathogenesis and aiding in the development of therapeutics. However, sequencing this highly contagious virus directly from samples is challenging in low- and middle-income countries lacking BSL-4 facilities. This study developed and optimized a culture-independent, high-throughput multiplex PCR-based third-generation sequencing protocol for NiV using the Oxford Nanopore Technology platform and a proposed bioinformatics pipeline to generate consensus genome sequences directly from environmental and clinical specimens. We amplified 12 NiV RT-PCR-positive specimens (11 clinical, one environmental) to produce 60 amplicons, each approximately 400 bp, covering the entire ~18.2 kb genome. Using a two-step reverse transcriptase PCR approach, libraries were prepared with a ligation sequencing kit. Raw sequence data were then analyzed using bioinformatics tools. A minimum of 10,000 total reads per sample provided a nearly complete coverage (>95%) of the NiV genome, even with low virus concentrations (Ct ≤ 32), with an average quality score of 10.2. The WGS of 12 NiV-positive samples achieved coverage between 95.71% (Ct 29.54) and 99.3% (Ct 22.34). The entire process, from RNA extraction to finished sequences, took only 24 h. We developed a portable, culture-independent, high-throughput sequencing workflow suitable for resource-limited settings, aiding in real-time monitoring, outbreak investigation, and detection of new NiV strains and genetic evolution.

**IMPORTANCE** The development of a culture-independent, high-throughput whole-genome sequencing (WGS) protocol for Nipah virus (NiV) using the Oxford Nanopore MinION technology marks a significant advancement in outbreak response, surveillance, and genomic analysis of NiV. NiV is an RG4 category C pathogen; working with the NiV virus is a deep concern of biosafety and biosecurity. It demands the development of biologically safe procedures to get genetic information. This protocol utilizes biologically safe samples that were collected into recommended lysis solution, multiplex PCR, and third-generation sequencing, effectively addressing challenges in sequencing NiV. This optimized workflow achieved over 95% genome coverage without the need for virus culture. It is a cost-effective, rapid, and efficient approach to the WGS of NiV, making it suitable for resource-limited settings like Bangladesh. The method enhances the capacity for outbreak investigations, epidemiological analyses, and monitoring virus, aiding in

**Peer Reviewer** Neta S. Zuckerman, Israel Ministry of Health, Ramat Gan, Israel

Address correspondence to Mohammed Ziaur Rahman, mzrahman@icddrb.org.

Md. Mahfuzur Rahman and Mojnu Miah contributed equally to this article. The order of names of the two first authors was determined alphabetically.

The authors declare no conflict of interest.

detecting emerging strains. This work contributes significantly to global pandemic preparedness and response efforts.

**KEYWORDS** Nipah virus, Oxford Nanopore MinION, culture independent, whole-genome sequencing, clinical and environmental specimen

Nipah virus (NiV) is a zoonotic virus with pandemic potential of the *Paramyxoviridae* family belonging to the genus *Henipavirus*, which causes severe neuro-invasive (encephalitis, meningitis) and respiratory diseases in humans and animals in Southeast Asia and the Western Pacific regions (1). It is classified as a biosafety level (BSL) 4 agent and a category C priority pathogen due to its extreme virulence and high fatality rate according to the National Institute of Allergy and Infectious Diseases (2, 3). It has repeatedly caused outbreaks annually between December and March in Bangladesh and India, with a case fatality rate exceeding 75% (4, 5).

The Nipah virus possesses a single-stranded, negative-sense RNA genome of approximately 18.2 kb in size with six structural and three non-structural genes (6). Based on disease progression and transmission, Nipah virus strains are divided into two categories: Nipah virus Malaysia (NiV-M) and Nipah virus Bangladesh (NiV-B) (7). Bangladeshi NiV strains have recently diverged into two key sub-lineages (i.e., NiV-BD 1 and NiV-BD 2). According to phylogenetic analyses, all strains from Bangladesh belong to the NiV-BD genotype and evolve at a rate of $4.64 \times 10^{-4}$ substitutions/site/year (8). This evolution monitoring is important for vaccine design and drug development. The ability of NiV to evolve quickly further emphasizes the need for constant monitoring and surveillance, especially at the genomic level to track the emergence of novel strains. To address these issues, the whole-genome sequencing of NiV can provide information on the identity and evolution of genetic variations and the origin of this virus. The NiV whole-genome sequence can also map conserved and mutable regions, which would provide valuable insights into the epidemiology of the virus and the development of effective treatments and vaccines. Moreover, the evolutionary analysis could be useful for the development of vaccines, new therapies, and prevention strategies. As of August 2024, a very limited number of NiV full-genome sequences ($n = 107$) are available in public repositories, such as the National Center for Biotechnology Information (NCBI).

A range of sequencing approaches is currently employed for Nipah virus genomic analysis, including metagenomic sequencing, targeted enrichment strategies, and PCR amplification-based methods (9). However, it is extremely challenging to extract and sequence the complete genome of NiV from environmental, clinical, and animal samples without virus culture due to the low viral load. For virus isolation, Biosafety Level (BSL) 4 confinement facilities are required, but these facilities are not widely available across the globe. Due to the high costs associated with traditional next-generation sequencing (NGS) platforms like Illumina and Ion Torrent, many research laboratories in resource-constrained countries, such as Bangladesh, have found them difficult to afford over the past decade. In contrast, the MinION platform, a third-generation nanopore-sequencing device, offers a more cost-effective alternative, making it particularly well-suited for use in viral sequencing studies within these laboratories. Large-scale whole-genome sequencing via metagenomic sequencing and targeted enrichment is excessively expensive and accompanied by inadequate coverage and depth (10). These two procedures require a sample with a higher viral load. Those limitations can be overcome by utilizing a PCR-based amplification method that simultaneously enriches and amplifies the target of interest (11). It is possible to achieve adequate coverage and depth for synthesizing NiV full-genome sequences using a tiled amplicon method and multiplex PCR at a reasonable cost. This strategy enables the assembly of whole genomes, even from samples with low viral loads or partially degraded viral RNA genomes (12, 13).

In this study, we describe the development of a nanopore-based culture-independent method for the efficient genome sequencing of the Nipah virus with Ct values

ranging from 25.52 to 37.72. Our approach involves a multiplex PCR amplification-based sequencing assay and a bioinformatics analysis method. To amplify the whole genomes of the Nipah virus, we synthesized 60 amplicons. These amplicons were subsequently sequenced using the portable Oxford Nanopore MinION MK1C device. The resulting sequence data from the MinION device were then processed using a newly developed bioinformatics workflow. This research offers important insights for future applications of culture-independent techniques in sequencing highly pathogenic viruses.

## MATERIALS AND METHODS

### Specimen collection

A total of 12 archived clinical and environmental samples (human throat swabs = eight, blood serum = one, human urine = one, human breast milk = one, and bat roost urine = one) with evidence of NiV RNA using real-time reverse transcription-polymerase chain reaction (RT-PCR) were subjected to whole-genome sequencing using the Oxford Nanopore MinION technology. Biological specimens, including throat swabs, serum, urine, and breast milk from patients suspected of the Nipah virus infection, were collected using 3 mL cryotubes containing NUCLISENS easyMAG lysis buffer (NucliSENS easyMag, bioMerieux, Inc., Rodolphe St., Durham, NC, USA). These procedures were carried out by trained medical technologists, adhering to strict biosafety and biosecurity guidelines.

In parallel, bat-pooled urine samples were collected from roosting sites in the Joypurhat District. To collect the bat urine, polyethylene tarps (6 ft × 4 ft) were placed beneath each roost between 12:00 and 4:00 AM. Samples were retrieved between 5:00 and 6:00 AM using sterile syringes and stored in 50 mL Falcon tubes. The urine was then aliquoted in duplicate into viral transport media (VTM) and TRIzol reagent (Invitrogen, CA, USA) (0.3 mL urine mixed with 0.9 mL VTM/TRIzol). All specimens were immediately transported to the One Health Laboratory at the International Centre for Diarrheal Disease Research, Bangladesh (icddr,b) in liquid nitrogen, maintaining cold-chain requirements for viral preservation, and stored at −80°C for long-term preservation and subsequent diagnostic testing. These samples were collected as part of the "Nipah Virus Transmission in Bangladesh" (PR-2005-026) study and the "Study of Nipah virus dynamics and genetics in its bat reservoir and of human exposure to NiV across Bangladesh to understand patterns of human outbreaks" (PR-21085).

### Viral RNA extraction and qRT-PCR

Viral RNA was extracted from 140 µL of human samples using the QIAamp Viral RNA Mini Kit (Qiagen, Hilden, Germany). For the bat roost urine samples preserved in TRIzol reagent, 200 µL was processed using the Direct-zol RNA Miniprep Kit (Zymo Research, CA, USA). In both cases, the RNA extractions were carried out following the respective manufacturers' protocols. Detection of Nipah virus RNA in the RNA extract was carried out using RT-qPCR, with the TaqMan PCR assay targeting the NiV N gene through specific primers and probes (14). One-step RT-qPCR assays were conducted using the iTaq universal probes and the One-step Reaction Mix Kit (Bio-Rad Laboratories, CA, USA) on a CFX Opus Real-time PCR System (Bio-Rad Laboratories, CA, USA)

### cDNA preparation and multiplex ARTIC PCR to amplify viral genome sequences

The extracted RNA was converted to complementary DNA (cDNA) without a freeze–thaw cycle to prevent possible degradation of RNA by using the LunaScript RT SuperMix Kit (New England Biolabs, Inc.). Following synthesis, cDNA was stored at 4°C for a short period while waiting for multiplex ARTIC PCR and qPCR. In this study, we used two pools of primers designed by the ARTIC Network (15) but modified by our lab experts. Primer sequences (modified bases are indicated in bold red color) are shown in Table

S1. For each sample, two pools of PCR reactions were prepared, with pools 1 and 2 each containing 60 primer sets. These primer pools were used to generate overlapping amplicons, minimizing interference during the production of short overlapping products. This tiling amplicon approach ensured complete PCR amplicon coverage of the genome. For the PCR reaction, we used the Q5 High-fidelity DNA Polymerase from New England Biolabs (NEB). The amplicon size of the ARTIC primer panel for the NiV was approximately 400 bp; however, due to overlapping target sites, some amplicons exceeded this intended length. To visualize the bands, the PCR products from pools 1 and 2 were loaded onto the 1.5% agarose gel and imaged using the Gel Doc XR + Gel Documentation System (Bio-Rad Laboratories, Inc., Hercules, CA, USA).

## Pooling and purification of ARTIC PCR products

The PCR products from primer pools 1 and 2 were combined into a 25 µL total volume and subsequently purified using 0.64× AMPure XP beads (Beckman Coulter, A63881, Indianapolis, USA) according to the amplicon clean-up protocol. The resulting pellet was dissolved in 20 µL of nuclease-free water. A 1 µL aliquot of this solution was quantified using the Qubit dsDNA HS Assay Kit (Thermo Fisher, California, USA) following the manufacturer's guidelines in preparation for library construction.

## Quantification of pooled and purified ARTIC PCR products

This strategy uses the concentration of the no template control (NTC) typically containing primers and artefacts to assess the amplicon quality. For NTC with a concentration of 27.2 ng/µL, we defined three grading criteria: grade one for amplicons ≥ 62 ng/µL; grade two for those between 28 and 62 ng/µL; and grade three for amplicons < 28 ng/µL. Samples of the same grade are pooled for sequencing in the same run, improving the efficiency and accuracy of the sequencing process. However, for downstream processes, all grade three samples are excluded. It is important to note that the negative controls are included in all the sequencing runs.

## Preparation of DNA ends for barcoding

For the preparation of nanopore sequencing libraries, we used the ligation sequencing kit (SQK-LSK109) and assigned each library a unique barcode using the native barcodes (EXP-NBD104). After purifying the multiplex amplicons, an end-prep reaction was performed using the NEBNext Ultra II End Repair/dA-Tailing Module (New England BioLabs, MA, USA) with certain modifications. The reaction was set up by combining 1.75 µL of Ultra II End Prep Reaction Buffer, 0.75 µL of Ultra II End Prep Enzyme Mix, and 5 µL of $H_2O$ with 2.5 µL of each multiplex amplicon, resulting in a final volume of 10 µL. The mixture was incubated in a thermocycler at 20°C for 15 min, followed by 65°C for 15 min. The resulting end-prepped amplicons were subsequently used for the native barcode ligation step. In this step, 2.5 µL of end-prepped amplicons was combined with 5.75 µL of Blunt/TA Ligase Master Mix, 0.5 µL of $H_2O$, and 1.25 µL of each native barcode. The reaction mixture was incubated at room temperature for 20 min, followed by a 10 min incubation at 65°C.

## DNA library sample barcoding and adapter ligation

A set of 12 amplicons with unique barcodes was combined into a single pool. The pooled amplicons were purified using 0.64× AMPure XP beads from Beckman Coulter, CA. The purification process involved two successive 250 µL washes with short fragment buffer (SFB) and an additional wash with freshly prepared 80% alcohol. The barcoded library was then eluted in 30 µL of elution buffer (EB) and quantified using the Qubit 1× Double-stranded DNA (dsDNA) High-sensitivity Assay Kit (Thermo Fisher Scientific, USA) with a Qubit 4 fluorometer (Thermo Fisher Scientific, USA). Following quantification, the library was subjected to adapter ligation for sequencing.

## Loading and running the MinION

A mixture of 10 µL of 5× NEBNext quick ligation reaction buffer, 5 µL of Quick T4 DNA ligase, and 5 µL of ONT Adapter Mix II (AMII) was combined with 30 µL of the barcoded library to make a final reaction volume of 50 µL and incubated at room temperature for 20 min. Subsequently, the library was purified using 1× (1:1) AMPure XP beads, which involved two consecutive 250 µL washes with SFB. The resulting purified library was eluted in 15 µL of the EB buffer from ONT. To determine the quantity of the final library, it was quantified using the Qubit dsDNA HS Assay Kit (Thermo Fisher) and a Qubit 4 fluorometer. Approximately 60 ng (12 samples) of the final library was loaded onto the FLO-MIN106D (R9.4.1) flow cell on an Oxford Nanopore MinION MK 1C platform. The library was then subjected to a 12 h sequencing run.

## Bioinformatics workflow for the MinION pipeline

The primary fast5 signal data were subjected to real-time base-calling to fastq files using Guppy v.4.3.4 (Oxford Nanopore Technologies). This particular procedure was executed within the MinKNOW software operating in a fast base-calling mode. The resulting fastq reads were subsequently demultiplexed and categorized into distinct barcodes employing Guppy v.4.3.4. To ensure the integrity and quality of the data, the demultiplexed reads underwent quality checks within the EPI2ME WIMP workflow. After obtaining the sequence summary file, pycoQC tools (16) were employed for quality control assessments. Subsequent to this, we carried out trimming of barcode and adapter sequences through Porechop v.0.2.3 (17), segregating chimeric reads into individual reads. Following this, the trimmed fastq reads for each sample were aligned against the Nipah virus reference sequence (AY988601). This alignment was carried out using the minimap2 pairwise aligner with the "ax map-ont" setting (18). The assembled file was then polished, and the consensus sequence was called via the mpileup command in SAMtools and BCFtools (v.1.5.0) (19). Ambiguities and indels in homopolymer regions were corrected manually based on the Bangladeshi reference genome (AY988601) using the Integrative Genomics Viewer (IGV) v.2.12.2 tool to map coverage of the alignments and checked by utilizing the qualimap tool v.2.2.2. Finally, consensus FASTA files were generated from the consensus fastq files via the seqtk tool (20). We plotted the coverage and further evaluation of the read quality using NanoPlot (v1.25.0) tools to receive mean read quality and read length histograms. The consensus sequences generated by the reference-based assembly were validated by BLAST searches to confirm Nipah virus genotypes and detect nucleotide identity. For each sample, all the mapped reads were analyzed. The complete sequence data are deposited in the SRI database under the NCBI.

## Phylogenetic analysis

A phylogenetic tree was constructed using all successfully sequenced Nipah virus genomes. Additionally, 74 whole-genome sequences (~18,200 bp) of the Nipah virus strains sequenced between 1999 and 2023 and curated from the NCBI GenBank database were included in the analysis. Multiple sequence alignment was performed using BioEdit (version 7.2.5.0) to ensure an accurate comparison of the selected sequences. The phylogenetic tree was generated using the maximum likelihood method in MEGA (version 11.0.13), applying the T93 + G + I substitution model (21), as determined by ModelFinder (22). The robustness of the tree was assessed through phylogenetic bootstrapping with 1,000 replicates. SNPs between the newly sequenced whole genome of NiV strains and the Bangladeshi reference genome (AY988601) were identified using the Python-based tool, SNP-site (23).

## RESULTS

A total of 12 archived NiV-positive samples (clinical and environmental) were selected for the development and optimization of our Nanopore sequencing protocol, displaying

Ct values ranging from 22.34 to 37.78. We carried out a complete genome sequencing of 12 RT-PCR-confirmed Nipah virus-positive samples using multiplex-PCR with modified ARTIC primer sets by Oxford Nanopore Technology (ONT). This methodology/workflow applies to animal, human, and environmental samples. The entire workflow to generate a complete genome sequence (Fig. 1) of the RT-qPCR-positive Nipah virus is performed in the following step-by-step procedure:

We constructed a DNA library by combining 12 uniquely barcoded ARTIC PCR products, each generated from one of the 12 RT-PCR-confirmed Nipah virus-positive samples. A 12 h sequencing run was performed using the pooled DNA library on the FLO-MIN106D (R9.4.1) flow cell of the Oxford Nanopore MinION MK 1C platform. This sequencing approach successfully generated complete genomes for nine out of 12 Nipah virus-positive samples, with coverage ranging from 98.01 (Ct 29.26 ± 8.54) to 99.79% (Ct 22.34). Among the remaining three samples, two with high Ct values (30 ≤ Ct ≤ 35) yielded partial genome coverage between 70.02% and 86.05%. However, for samples with Ct ≥ 35, near-complete genome coverage could not be achieved (Table 1; Fig. 2). This sequencing run utilized a new flow cell containing 700 available pores (Fig. 3).

"In total, 243,040 (138,764,876 bases) raw reads were generated, of which more than 95% (237,106 reads; 116,996,861 bases) passed quality control assessments. The average quality score of the reads was 10.32, ranging from 9.9 to 11 (Fig. S2) and resulting in 237,106 reads available for analysis. The average read length was reported as 554 bp (Fig. 4; Fig. S1), which was slightly larger than the expected amplicon size (bp) due to minor adjustments to the primer sets. In our run, we did not get any NiV-specific mapped reads in the NTC (no template control) sample, which underscores the reliability of the workflow and the integrity of the data generated for the study (Table 1).

The N50 value representing the sequence length, at which 50% of the total sequenced bases are found in reads of this length or longer, exceeded 615 nt [range between 939nt for FP0300923 (breast milk) and 431 nt for DMCH00123 (TS)]. Nine out of the 12 Nipah virus strains exhibited genomic coverage exceeding 98% (Fig. 5A). The proportion of sequencing reads that failed to pass the quality control filtering step for these samples ranged from 0.7 to 4.29%. The coverage of the reference Nipah virus genome across the sequenced samples exhibited an inverse correlation with the Ct value, whereby an increase in the Ct value corresponded to a reduction in the mean genomic coverage (Fig. 5B). Specifically, FP0300923 (breast milk), which had a Ct value of 22.34, demonstrated the highest mean coverage at 1,157-fold, whereas DMCH00123 (Ct value of 37.72) displayed the lowest mean coverage, measuring less than onefold (Table 1).

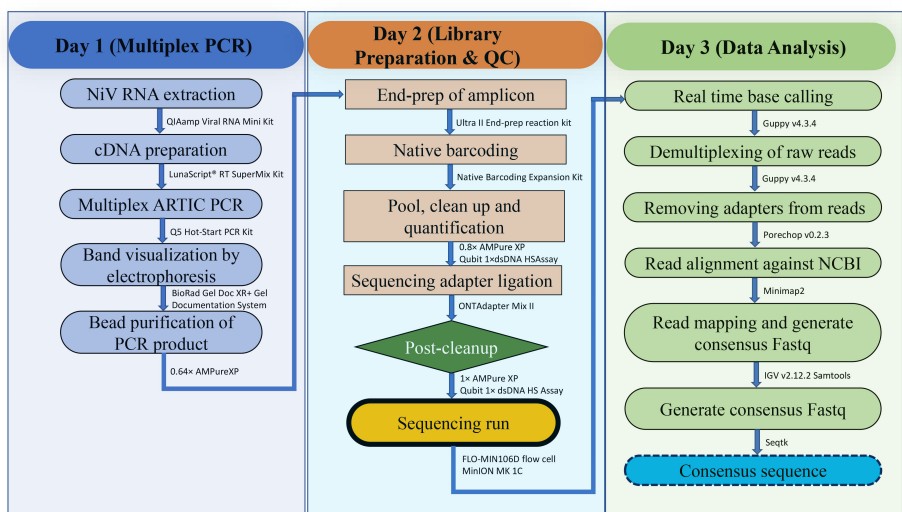

**FIG 1** Schematic illustration of the MinION working pipeline of full-length Nipah virus genome sequencing.

**TABLE 1** Summary of the sequencing data obtained from MinION for each sample with name and sample type[a]

| SL | Sample | Sample type | Clinical/Env | Ct value | Genotype | Total reads obtained (×1,000) | Mapped reads (×1,000) | Mapped% | Mean genomic coverage | Genomic coverage | Highest nt identity | GenBank ID |
|---|---|---|---|---|---|---|---|---|---|---|---|---|
| 1 | RS0644723 | Throat swab | Clinical | 27.16 | NiV-BD | 24959 | 24305 | 97.38% | 728 | 99.12% | 99.76% | PP981676 |
| 2 | NINS-00123 | Throat swab | Clinical | 29.54 | NiV-BD | 16522 | 15813 | 95.71% | 463 | 99.11% | 99.29% | PP981677 |
| 3 | NINS-00123 | Urine | Clinical | 32.25 | NiV-BD | 5406 | 5096 | 94.27 | 117 | 99.11% | 99.59 | PQ368168 |
| 4 | FP0298523 | Throat swab | Clinical | 26.02 | NiV-BD | 27720 | 27285 | 98.43% | 789 | 99.12% | 99.74% | PP981678 |
| 5 | FP0298923 | Throat swab | Clinical | 25.52 | NiV-BD | 29098 | 28772 | 98.88% | 876 | 99.20% | 99.79% | PP981679 |
| 6 | RS0649923 | Throat swab | Clinical | 32.86 | NiV-BD | 3257 | 2,248 | 69.02% | 49 | 70.02% | 99.74% | PP981679 |
| 7 | DMCH-00123 | Throat swab | Clinical | 37.72 | NiV-BD | 1064 | 10 | 0.94% | 0.29 | 27% | 99.59% | Not submitted to the NCBI |
| 8 | FP0300923 | Throat swab | Clinical | 27.91 | NiV-BD | 32648 | 32276 | 98.86% | 692 | 99.11% | 99.69% | PP981681 |
| 9 | FP0300923 | Breast milk | Clinical | 22.34 | NiV-BD | 35904 | 35561 | 99.04% | 1157 | 99.21% | 99.79% | PQ368169 |
| 10 | RS0652823 | Serum | Clinical | 33.2 | NiV-BD | 8810 | 8748 | 99.30% | 184 | 98.03% | 99.21% | PP981682 |
| 11 | FP302023 | Throat swab | Clinical | 26.88 | NiV-BD | 38233 | 37787 | 98.83% | 849 | 98.01% | 99.68% | PP981683 |
| 12 | J106 | Bat roost urine | Environmental | 32.7 | NiV-BD | 19840 | 19469 | 98.13% | 730 | 86.05% | 99.26% | PQ368170 |
| 13 | NTC | Negative template control | Control | N/A[b] | N/A | 71 | 0 | 0% | 0 | 0% | N/A | N/A |

[a]The number of nucleotide differences, mean coverage, and mapped reads are shown for each sample. The consensus identity is calculated for all segments in percentage (%).
[b]"N/A" stands for Not Applicable.

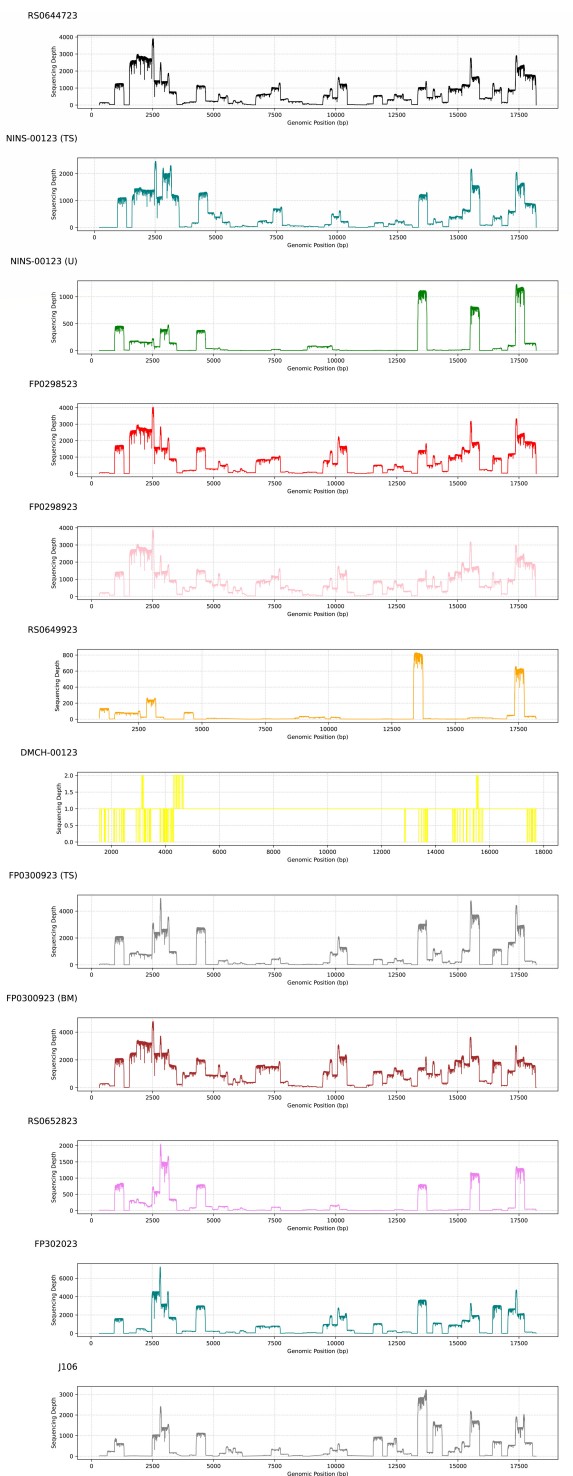

**FIG 2** Coverage and sequencing depth plots obtained for the 12 sequenced NiV samples across the genomes.

Among the 12 Nipah-positive samples identified via RT-qPCR, which had Ct values ranging from 22.34 to 37.72, a successful genome assembly (utilizing both *de novo* and map-to-reference approaches) was achieved for seven samples with Ct values ≤ 30. These samples exhibited high mean genomic coverage (≥400-fold) and achieved identities of more than 98% with reference sequences across all target genomes.

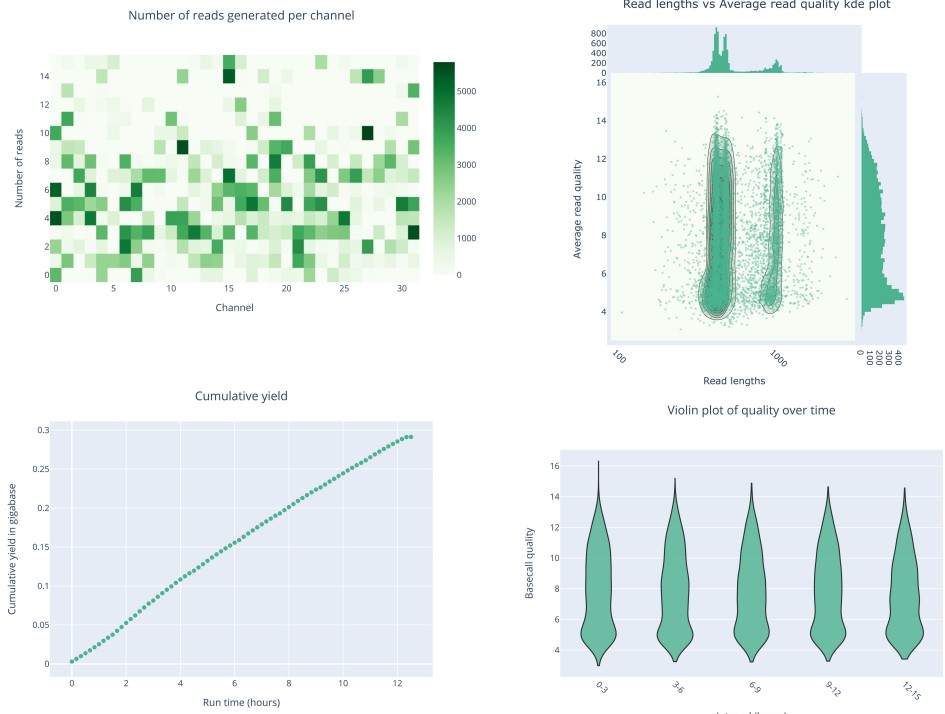

**FIG 3** Quality assessment of the Oxford Nanopore sequencing reads obtained from PCR amplicons of the NiV whole genome. The graphs display different quality metrics for the combined reads from four sequencing runs: (A) the number of reads per nanopore channel; (B) read length compared to the average read quality; (C) cumulative yield (in gigabases) over-run time; and (D) basecall quality over run time. These plots were created using NanoPlot (v1.42.0).

Full-genome sequences were obtained from all samples with Ct values below 30 in the RT-qPCR.

For the samples with Ct values between 30 and 35 ($n = 4$), the outcomes varied significantly, with genomic coverage ranging from 69.02 to 99.3%. Although two samples (RS0652823 and J106) with Ct values of 33.20 and 32.70 derived from human blood serum and bat roost urine displayed a notably high genomic coverage of 99.3 and 98.13%, respectively, the number of specific reads was 8,810 and 19,469 and a mean genomic coverage of 184, both of which were lower than the samples with Ct values lower than 30. In contrast, the sample with a Ct value of 37.72 exhibited only a few specific reads (10 mapped reads). Furthermore, the mean reading quality (Q) of these samples was lower compared to samples with lower Ct values (Fig. S2). Random subsampling of reads from high-coverage samples revealed that 10,000 reads are sufficient to achieve ≥98% genome coverage for high-quality Nipah virus samples.

Phylogenetic analysis using the maximum likelihood method on 85 whole-genome NiV sequences demonstrated that Bangladeshi strains form a phylogenetically distinct lineage, further dividing into two sub-lineages. Of the 12 newly sequenced genomes, 11 were included in this analysis, all clustering within the NiV-BD2 sub-lineage (Fig. 6A).

Single nucleotide polymorphisms (SNPs) were identified in 11 out of 12 NiV genomes by comparing them to the Bangladeshi NiV reference sequence (accession no.: AY988601.1). Among the nine NiV strains with genome coverage exceeding 98%, the number of SNPs ranged from 164 to 206, with a frequency of around 1% (Fig. 6C). The strain NINS-00123 (U) exhibited the highest SNP count (206), while RS0652823 had the lowest (164) (Fig. 6B and C). In contrast, due to lower genome coverage, RP0649923 and J106 contained fewer SNPs, with counts of 116 and 122, respectively. Additionally, no insertion or deletion mutations were identified in the 12 newly sequenced NiV genomes.

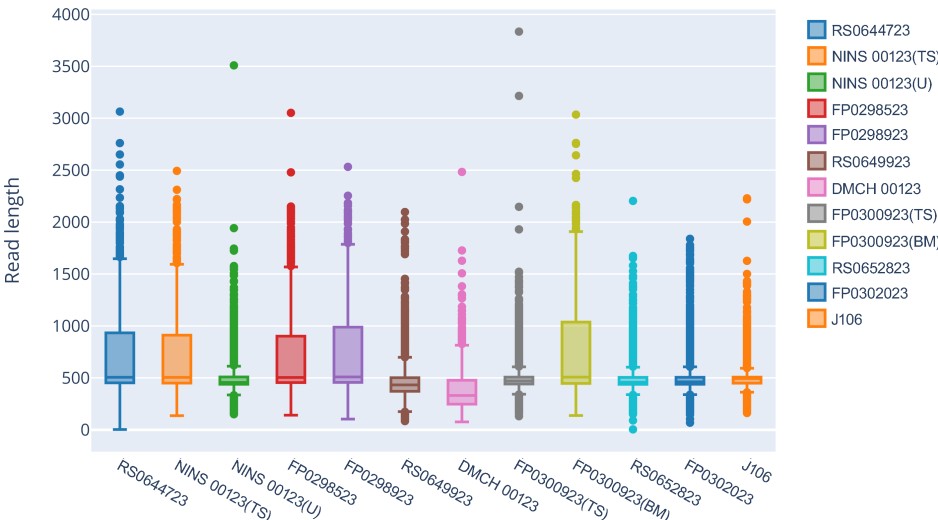

**FIG 4** Box plot of the read length for Nipah virus whole genome sequencing from 12 samples with an Oxford Nanopore MinION flow cell.

## DISCUSSION

The Nipah virus has posed a significant public health challenge in Bangladesh since its discovery in 2001, with intermittent outbreaks. It is crucial to quickly identify and characterize the virus during NiV outbreaks to implement early intervention measures like targeted containment strategies, contact tracing, and future vaccination campaigns. Real-time genomic information is essential for public health authorities to make informed decisions to curb the spread of the virus, thereby reducing its impact on human health. In resource-limited settings like Bangladesh, where there may be a lack of laboratory infrastructure, nanopore sequencing provides a transformative tool to enhance diagnostic capabilities. Unlike traditional methods that require specialized equipment and lengthy processes, nanopore sequencing is portable, scalable, and adaptable to field conditions. This technology can be deployed in remote areas to rapidly diagnose NiV infections, differentiate viral strains, and track changes in viral genomes over time.

This study aimed to evaluate the effectiveness of the MinION nanopore sequencer developed by Oxford Nanopore Technologies for sequencing the complete genome of the Nipah virus using a single flow cell. Current methods for identifying and characterizing new NiVs rely on Sanger and next-generation sequencing. While Sanger sequencing is highly accurate, it is low-throughput and not suitable for large-scale sequencing efforts. On the contrary, NGS provides higher throughput but can be time-consuming and often generates shorter sequence reads, typically only a few hundred nucleotides in length. These short reads can complicate the assembly process and lead to fragmented or incomplete genomes. Additionally, viral genome reconstruction using metagenomic data has the potential to produce chimeric assemblies, further complicating accurate genomic analyses. By contrast, ONT's nanopore sequencing technology, which is capable of producing long reads, offers a promising solution to overcome these limitations. It allows for the rapid and comprehensive sequencing of full-length viral genomes, providing a more efficient and precise method for Nipah virus genome assembly and characterization.

Recovering viral RNA from bat roost urine and conducting genome sequencing can be challenging due to several factors. The low viral load in such samples often means that only small amounts of viral RNA are present. Additionally, environmental conditions like temperature, humidity, and UV light, as well as the presence of RNases, can lead to rapid degradation of RNA. The presence of substances, such as dirt, chemicals,

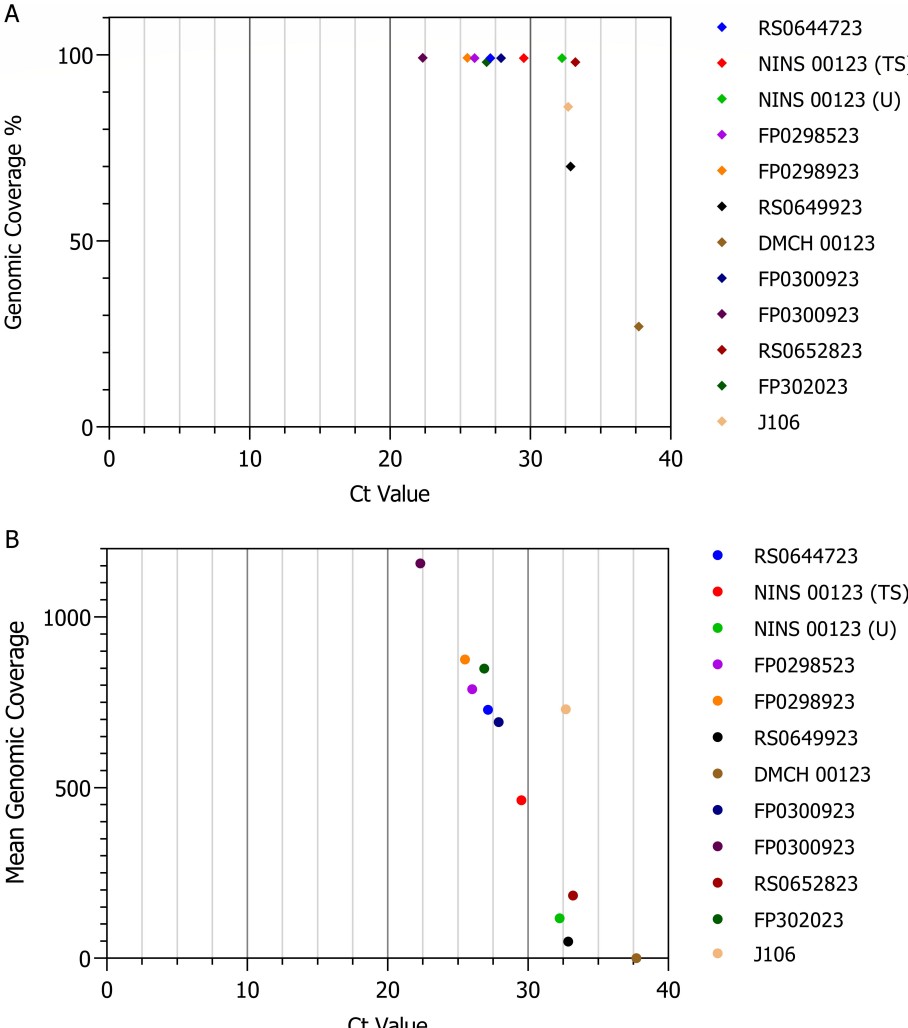

**FIG 5** Relationship between Ct values and genomic coverage. (A) Scatter plot showing the percentage of genomic coverage as a function of the Ct value for 12 NiV samples. (B) Scatter plot depicting the mean genomic coverage across Ct values. Each data point represents a unique sample, color-coded according to the legend on the right. Higher Ct values are generally associated with a lower genomic coverage.

and organic matter, in urine can act as inhibitors during RNA extraction and PCR, further complicating the process. Furthermore, contamination with genomic material from the complex microbial communities in the environment can make it difficult to isolate and accurately sequence the target viral RNA. Although culturing the Nipah virus for genetic analysis is a potential method, its high pathogenicity and the need for BSL-4 facilities make this approach highly challenging and impractical in most settings. To overcome these challenges, we employed a PCR amplicon nanopore sequencing approach. We implemented an RNA extraction protocol that includes DNase I enzyme treatment as part of the manufacturer's protocol for the Direct-zol RNA Miniprep Kit to selectively degrade host and bacterial DNA during the extraction process for environmental samples. Bat roost urine samples were aliquoted in TRIzol reagent in the field before RNA extraction. This step denatures all protein contaminants and selectively separates RNA from other contaminants in the aqueous phase, increasing the efficiency of RNA recovery. The extraction protocol also incorporates DNase I enzyme treatment to degrade any environmental DNA, which enhances the efficiency of RNA extraction, and the extraction strategy involves DNase I enzyme treatment to degrade environmental DNA, and these increase the efficiency of the RNA extraction. Additionally, TRIzol

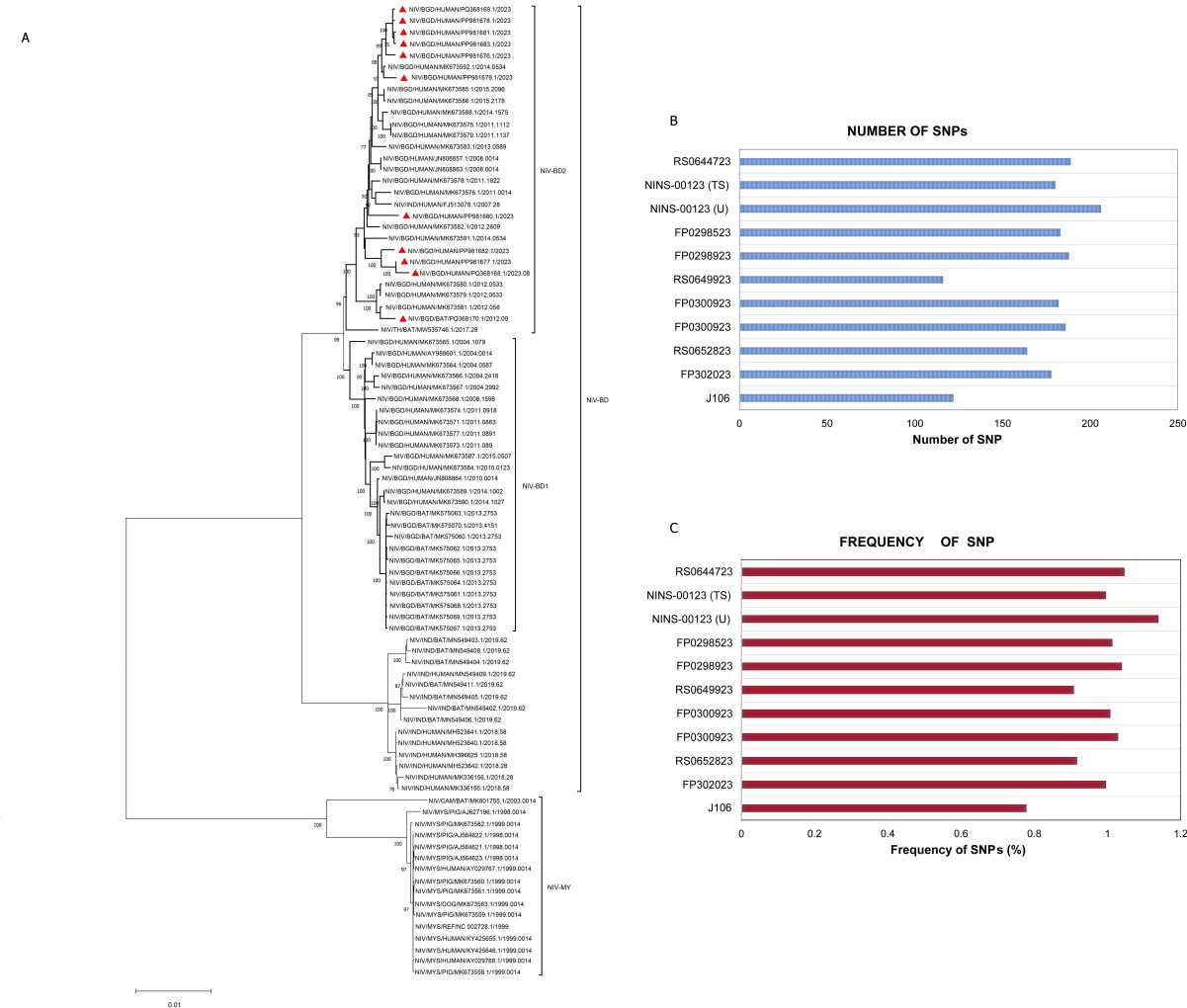

**FIG 6** Maximum likelihood phylogenetic tree of the whole genome sequences of Nipah virus (A). Sequences (red triangle) generated in this study clustered in the NiV-BD2 sub-lineage. (B) Number of SNPs in the newly sequenced NiV genome; and (C) frequency of SNPs in the genome of the newly sequenced 11 NiV strains.

reagent treatment further reduces RNA degradation caused by environmental RNases. Transporting the samples from the field to the laboratory in liquid nitrogen helps prevent RNA degradation due to environmental conditions,such as temperature, humidity, and UV light. Amplicon sequencing was used to amplify the target NiV genome from cDNA prepared from RNA extracted from NiV-positive samples using the efficient multiplex ARTIC PCR system containing two separate pools of primers, each with a non-overlapping 60-primer set.

Considering these advantages, we have successfully obtained the whole-genome sequences of the Nipah virus using our proposed high-throughput MinION sequencing protocol. This approach is versatile and can be applied in both field and clinical environments. We have sequenced 12 samples with 12 h of sequencing run, and this protocol can accommodate up to 48 samples at a time. The subsequent data analysis can be conducted in a cost-effective manner within the specified timeframe. The estimated cost of our native barcode-based sequencing protocol is approximately $30 per sample (24), making it a highly economical strategy. Additionally, the native barcoding protocol provides a swift multiplexing approach that eliminates the need for additional PCR amplification steps, streamlining the sequencing process. This enables the investigation of Nipah outbreaks and the real-time monitoring of genetic changes and new virus

strains. Nucleic acid extraction to sequence data acquisition take almost 3 days, whereas the nanopore sequencing run takes around 12 h.

Furthermore, our study indicates that samples with Ct values (in RT-qPCR for the N gene) below 30 exhibit a good sequencing performance when utilizing ARTIC PCR as a preamplification step for MinION sequencing. These Ct values demonstrated an inverse relationship with the sequencing quality in our study. Specifically, the sample group with a Ct value below 30 (Fig. 4) has a higher number of sequence reads with lengths exceeding 500 bp, and the N50 values for these seven samples are above 500 (Table 1). As a result, these samples have a higher mean genomic coverage (Fig. 5B). Additionally, samples with lower Ct values exhibit higher genomic coverage of the reference NiV genome (Table 1; Fig. 5A). In addition to quantitative benchmarks, sample quality is crucial, as the time and storage conditions of RNA samples significantly affect nucleic acid quality, especially in field samples where maintaining the cold chain can be challenging.

Our findings demonstrate that, with good sample quality, ARTIC PCR amplification and MinION sequencing can yield nearly complete genome sequences of the virus in most cases. These sequences achieve high levels of identity (98–99%) when compared to reference sequences in public databases. The protocol successfully sequenced near-complete genomes of the Nipah virus from various biological sources, including human throat swabs, blood serum, breast milk, urine, and bat roost urine. Notably, this protocol achieved 88.13% genomic coverage of the NiV genome from bat roost urine, representing a significant advancement over previous methods that faced challenges with low viral load, environmental contaminants, and RNase interference. The protocol was specifically optimized to improve RNA recovery and remove environmental contaminants. Furthermore, ARTIC PCR amplification was utilized to boost the copy number, addressing the issue of low initial viral load in bat urine.

A key modification in our approach was the optimization of ARTIC primers to enhance genome recovery for the Nipah virus (Table S1). While ARTIC protocols have been successfully applied to other viral pathogens, such as severe acute respiratory syndrome coronavirus 2, influenza, dengue, and Zika and Ebola viruses, the standard primer sets often require pathogen-specific adjustments to maximize coverage. Our modified primers were designed to improve amplicon balance, reduce dropout regions, and ensure consistent amplification across the Nipah virus genome. These modifications contributed to achieving 98–99% genomic coverage with a mean coverage depth ranging from 400 to 1,157 for samples with Ct values up to 32, demonstrating the effectiveness of our tailored approach. This modified primer set improved sequencing efficiency and genome completeness, particularly from challenging samples, such as bat roost urine. While ARTIC sequencing has been widely applied across viral genomes, pathogen-specific optimization remains crucial to overcoming challenges, such as low viral load and environmental interference.

Since the initial identification of the NiV in Malaysia in 1999, phylogenetic analyses have delineated two major genotypes: NiV-Malaysia (NiV-MYS) and NiV-Bangladesh (NiV-BD). Comparative genomic analyses indicate that these genotypes differ by approximately 8–10% at the nucleotide level (14). Moreover, similarity matrix analyses of NiV strains from Bangladesh collected between 2004 and 2023 demonstrate a high degree of genetic conservation, with nucleotide identities ranging from 98 to 100% relative to the Bangladeshi reference strain (AY988601) (8). In the present study, we identified approximately 200 SNPs in the 11 newly sequenced samples when compared to the 2004 Bangladeshi reference strain, suggesting a relatively low mutational rate for NiV. Given this observed rate, the current primer sets are expected to remain effective for an extended period. However, ongoing genomic surveillance is imperative, particularly for monitoring mutations in regions targeted by these primers. Furthermore, our findings underscore the robustness and adaptability of the ARTIC sequencing approach, providing a valuable framework for its application in the genomic surveillance

of emerging and re-emerging viral pathogens, particularly those necessitating high sensitivity for field and clinical samples.

The major advantages of ONT sequencing in this study are its affordability and the potential for application in resource-limited settings, including field conditions. While ONT is often recognized for generating long reads, the amplicons used in our study are approximately 400 bp in length and do not encompass the entire viral genome. As such, the main benefit of ONT in this context lies in its cost-effectiveness. We applied this strategy to investigate Nipah outbreaks in Bangladesh, showcasing its effectiveness in promptly investigating outbreaks in a cost-effective setting. Another advantage is the ability to sequence clinical samples directly, bypassing the need for virus culture. Furthermore, this method allows for sequencing NiV at low concentrations [cycle threshold (Ct) values ≤ 33].

Timely and accurate sequencing is crucial for effective surveillance and informed vaccine selection, ensuring that interventions remain relevant to the evolving viral strains. The proposed protocol is well-suited for Nipah virus surveillance in outbreak-prone countries like Bangladesh, where sequencing facilities are limited. It enables the identification of pandemic potential agents and the selection of vaccine strains from diverse sample types, including humans, bats, and other potential natural reservoirs, all within a cost-effective framework.

## Conclusions

Here, we have introduced whole-genome sequencing techniques for the Nipah virus using the Oxford Nanopore MinION platform with NiV-specific PCR amplicons in a two-step PCR. This assay successfully generates whole-genome sequences from animal, clinical, and environmental specimens of all Bangladeshi NiV genotypes, providing good genomic coverage. These cost-efficient and rapid approaches can be used for in-depth studies of NiV epidemiology, virus evolution, and identification of transmission dynamics during potential outbreaks and provide the foundation for designing novel intervention strategies specifically tailored for lower middle-income countries like Bangladesh.

### ACKNOWLEDGMENTS

The Nipah virus sequencing methodology is a part of the "Characterizing the epidemiological diversity of Nipah strains from Bangladesh" study, "Nipah Virus Transmission in Bangladesh" protocol (PR-2005-026), and "Study of Nipah virus (NiV) dynamics and genetics in its bat reservoir and of human exposure to NiV across Bangladesh to understand patterns of human outbreaks" funded by the Coalition for Epidemic Preparedness Innovations (CEPI), Centers for Disease Control and Prevention (CDC), and NIH National Institute of Allergy and Infectious Diseases (NIAID) NIH R01 award 1U01AI153420. The icddr,b acknowledges with gratitude the commitment of the CEPI, CDC, and National Institutes of Health (NIH) to their research efforts. The icddr,b is also grateful to the governments of Bangladesh and Canada for providing core/unrestricted support.

This research study was funded by the CEPI, the CDC, and the NIH NIAID NIH R01 award 1U01AI153420. We gratefully acknowledge our core donors, the governments of Bangladesh and Canada, for their support and commitment to icddr,b's research efforts.

The authors declare that there is no conflict of interest. The findings and conclusions in this manuscript are those of the authors and do not necessarily reflect the views of the US CDC, CEPI, EcoHealth Alliance or NIH.

### AUTHOR AFFILIATIONS

[1]Infectious Diseases Division (IDD), icddr,b, Dhaka, Bangladesh
[2]Department of Microbiology, Jashore University of Science and Technology, Jashore, Bangladesh

[3]Institute of Epidemiology, Disease Control and Research (IEDCR), Dhaka, Bangladesh

[4]EcoHealth Alliance, New York, New York, USA

[5]Viral Special Pathogens Branch, Division of High-Consequence Pathogens and Pathology, National Center for Emerging and Zoonotic Infectious Diseases, Centers for Disease Control and Prevention, Atlanta, Georgia, USA

## PRESENT ADDRESS

Mojnu Miah, Department of Biochemistry and Molecular Biology, University of Arkansas for Medical Sciences, Little Rock, Arkansas, USA

Ariful Islam, Gulbali Institute, Charles Sturt University, Wagga Wagga, NSW, Australia

## AUTHOR ORCIDs

Mohammad Enayet Hossain  http://orcid.org/0000-0001-5511-1857
Samiur Rahim  http://orcid.org/0000-0003-1468-395X
Shannon L. M. Whitmer  http://orcid.org/0000-0002-0819-6736
Mohammed Ziaur Rahman  http://orcid.org/0000-0002-4103-4835

## AUTHOR CONTRIBUTIONS

Md. Mahfuzur Rahman, Data curation, Formal analysis, Investigation, Methodology, Software, Validation, Visualization, Writing – original draft, Writing – review and editing | Mojnu Miah, Data curation, Formal analysis, Investigation, Methodology, Software, Validation, Visualization, Writing – original draft, Writing – review and editing | Mohammad Enayet Hossain, Methodology, Resources, Supervision, Writing – review and editing | Samiur Rahim, Data curation, Methodology, Writing – review and editing | Sharmin Sultana, Resources, Writing – review and editing | Syed Moinuddin Satter, Project administration, Resources, Writing – review and editing | Ariful Islam, Resources, Writing – review and editing | Shannon L. M. Whitmer, Resources, Writing – review and editing | Jonathan H. Epstein, Resources, Writing – review and editing | Christina F. Spiropoulou, Funding acquisition, Resources, Writing – original draft | John D. Klena, Funding acquisition, Resources, Writing – review and editing | Tahmina Shirin, Resources, Writing – review and editing | Joel M. Montgomery, Funding acquisition, Resources, Writing – review and editing | Maria E. Kaczmarek, Resources, Supervision, Writing – review and editing | Mohammed Ziaur Rahman, Conceptualization, Formal analysis, Funding acquisition, Investigation, Methodology, Project administration, Resources, Software, Supervision, Writing – original draft, Writing – review and editing | Iqbal Kabir Jahid, Formal analysis, Supervision, Writing – review and editing

## DATA AVAILABILITY

All data sets generated or analyzed during this study are included in the manuscript and/or the supplementary. The sequence data of this study were deposited to Gen-Bank under accession numbers PP981676, PP981677, PQ368168, PP981678, PP981679, PP981680, PP981681, PQ368169, PP981682, PP981683, and PQ368170. The data from this study can be found at the NCBI under BioProject accession number PRJNA1163683. The raw fastq reads have been deposited in the Sequence Read Archive database under accession numbers SRR30760182 to SRR30760192. The genome sequence of the sample DMCH00123 is not submitted to the NCBI database due to low genome coverage (<50%); those data are available on request from the corresponding author.

## ETHICS APPROVAL

This study was reviewed and approved by the Research Review Committee (RRC) and Ethical Review Committee (ERC) of the International Centre for Diarrheal Disease Research, Bangladesh (icddr,b), under protocol no. PR-2005-026 and PR-21085.

## ADDITIONAL FILES

The following material is available online.

### Supplemental Material

**Figure S1 (Spectrum02492-24-s0001.pdf).** Read length by the average read quality for the Nipah virus whole-genome sequencing from 12 samples with an Oxford Nanopore MinION flow cell.

**Figure S2 (Spectrum02492-24-s0002.pdf).** Scatter plot showing mean read quality (Q) as a function of the Ct value for 12 NiV samples.

**Table S1 (Spectrum02492-24-s0003.pdf).** Original and modified ARTIC primer sets used in this study.

### Open Peer Review

**PEER REVIEW HISTORY (review-history.pdf).** An accounting of the reviewer comments and feedback.

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
