## [Reviewer comments · Microbiology Spectrum]

Microbiology Spectrum

Development of a Culture Independent Whole Genome Sequencing of Nipah Virus Using MinION Oxford Nanopore platform

Md Rahman, Mojnu Mioah, Mohammad Hossain, Samiur Rahim, Sharmin Sultana, Syed Satter, Ariful Islam, Shannon Whitmer, Jonathan Epstein, Christina Spiropoulou, John Klena, Tahmina Shirin, Joel Montgomery, Maria Kaczmarek, Mohammed Ziaur Rahman, and Iqbal Jahid

Corresponding Author(s): Mohammed Ziaur Rahman, International Centre for Diarrhoeal Diseases Research Bangladesh

Review Timeline:

Submission Date:	October 4, 2024
Editorial Decision:	January 30, 2025
Revision Received:	March 1, 2025
Accepted:	March 10, 2025

Editor: Alexander Bello

Reviewer(s): Disclosure of reviewer identity is with reference to reviewer comments included in decision letter(s). The following individuals involved in review of your submission have agreed to reveal their identity: Neta S Zuckerman (Reviewer #3)

Transaction Report:

DOI: <https://doi.org/10.1128/spectrum.02492-24>

Re: Spectrum02492-24 (Development of a Culture Independent Whole Genome Sequencing of Nipah Virus Using MinION Oxford Nanopore platform)

Dear Dr. Mohammed Ziaur Ziaur Rahman:

Thank you for the privilege of reviewing your work. Below you will find my comments, instructions from the Spectrum editorial office, and the reviewer comments.

Revision Guidelines

Sincerely,
Alexander Bello
Editor
Microbiology Spectrum

Reviewer #2 (Comments for the Author):

In this article by Rahman et al, the authors design a strategy to perform Nipah virus whole genome sequencing using the long-read sequencer, ONT. The movement towards the ONT platforms for pathogen surveillance is growing in popularity and aligns well with resource limited settings, like Bangladesh described here. Overall, the article shows the feasibility of the work however I believe the though following would significantly strengthen the impact and provide clarity:

Minor

- In Methods 2.3 it states: "we used two pools of primers designed by the ARTIC Network but modified by our lab experts" What was modified on these primers? How do they differ from the standard ARTIC Network sets?
- The third paragraph of the results states the average read length was 554bp. It would be helpful to know the expected average for the ARTIC primers to see how well these aligned between expected and acquired reads.
- Figure 3: There is no indication which samples are corresponding to each panel. Please change the titles to the sample names rather than using the same title over and over. Also the figure text (e.g. title) appear to be different sizes, please correct to keep all the same size
- Table 1: This is a bit confusing. A few columns are labeled as x1000. I am unsure it would be possible to get this sequencing depth or reads for this flow cell and the number of samples pooled (e.g. it reads that sample 1 got 728,000x coverage).
- Table 1: Rather than showing "Highest Nucleotide Identity" (or perhaps in addition to this column) I recommend you show genome coverage. This gives a false impression that sample 7, which only got 10 reads, still had >99% genome coverage.
- Did the NTC sample have any detection of Nipah virus reads (suggesting cross-contamination). Please put the NTC data/stats in Table 1
- ARTIC sequencing on ONT platforms isn't completely unique to this study. How do your results compare to other ARTIC protocols for other viral pathogens? Could expand in the discussion section
- ARTIC protocols make use of PCR products. How often do you predict new primer sets will need to be based on NiV mutational frequency?
- It would be really interesting to see what the minimum number of reads are required for a sufficient genome coverage. This is especially useful for ONT platforms where you can stop, flush the flow cell, and sequence again. You could do this by randomly sampling reads from the samples with high genome coverage and increase the number of sampled reads until you reach a genome coverage threshold (e.g. 98%). Then you can calculate the average number of reads needed.
- These appear to be archived samples. Were this originally sequenced with other platforms (i.e. Illumina)? If so, it would be interesting to compare the results between ONT and Illumina results.

Major

- In the discussion it states, "We implemented an extraction protocol involving DNase I enzyme treatment to selectively degrade host and bacterial DNA during the RNA extraction process for the clinical samples." There is no data presented regarding this approach and I did not see description of this in the methods section. I would strongly encourage that this data is included to show the efficiency of your extraction approach
- Generally speaking, the majority of the data is showing QC information for the run metrics, which is helpful to have. However, where this publication lacks is the strength of how the data generated is useful from a pathogen genomics perspective. For example the following would substantially increase the impact:
 - o How divergent are these viruses? How many SNPs/indels were detected?
 - o Phylogenetic trees would be helpful here to show the diversity of viral sequences captures
 - o Plotting the data showing genome coverage across the samples would be incredibly helpful. There likely are biased areas of the genome with greater depth, but there is no data showing this.

Reviewer #3 (Comments for the Author):

This study presents the development of a rapid, culture-independent workflow for whole genome sequencing of Nipah virus (NiV) using Oxford Nanopore technology. It addresses two critical challenges in resource-limited settings dealing with highly infectious viruses: (1) enabling safe sequencing directly from clinical samples without viral culturing, using amplicons that span the entire genome, and (2) providing a fast and cost-effective sequencing approach with an affordable device. While the authors did not design the primers themselves, they optimized their use and demonstrate their effectiveness across relevant clinical and environmental samples with varying viral loads. Notably, this study underscores the vital role of genome sequencing in public health, particularly in resource-constrained regions facing viral outbreaks, facilitating real-time outbreak monitoring, NiV transmission investigation, and tracking viral evolution.

Comments:

- In the introduction:

"...substitutions/site/year (8) while this evolution monitoring is important..."

change to: ...substitutions/site/year (8). This evolution monitoring is important

- In the introduction:

"In addition to addressing these issues, whole-genome sequencing..."

change to: To address these issues...

- In methods, 2.1 - a possible typo - "icddr,b" ?

- In methods, 2.3 - "we used two pools of primers designed by the ARTIC Network but modified by our lab experts". How / where were the primers modified and why?

- In methods, 2.8 - what was the reasoning behind choosing only 12 samples at a time to load onto the flow cell? What's the maximum number of positive samples that can be loaded into a single flow cell at a time?

- In methods, 2.8 - what was the reasoning for using an older-version flow-cell (R9.4.1) rather than the newer available versions (R10.4)?
- In methods, 2.9 - what's the reasoning behind analyzing only 10,000 reads per sample?
- In results, what's the reasoning behind running the device 12 hours (and not less/more)? Did you see that enough genomic coverage was achieved only after 12 hours? maybe time can be saved for lower Ct samples if the run stops before. I think you should add a figure with a graph that states the coverage of the genome as a function of sequencing time and maybe even the Ct of the sample. You can refer to this new figure from parts in your results section that you write about but not represent in a figure. E.g. this sentence: "The coverage of the reference Nipah virus genome across the sequenced samples exhibited an inverse correlation with the Ct value, whereby an increase in the Ct value corresponded to a reduction in mean coverage" should be shown in a figure. I would put this figure instead of Figure 2 and put figure 2 in the supplementary.
- In results, 12 samples were run but you say that only 9 complete genomes were achieved. The phrasing of this part is very confusing and should be rephrased.
- Table 1 - please add the genomic coverage % for each sample.
- Figure 3 - belongs in the supplementary, and is not referred to in the text.
- Figure 4 is not referred to in the results section.
- In results, it would be nice to add a figure in the supplementary for this sentence: "Furthermore, the mean reading quality (Q) of these samples was lower compared to samples with lower Ct values"
- In Discussion - ONT and NGS were already defined earlier in the manuscript.
- In Discussion - you mention that one of the main advantages of ONT is that it produces long reads, however this doesn't serve this paper, as the amplicons are 400bp in length (which is a bit longer than Illumina reads but still is not the whole genome). The main advantage of ONT in this case is that it's cheaper and can be applied in third world countries and "in the field" if necessary.
- In Discussion - this sentence - "The limited presence of viruses in clinical samples, along with host nucleic acid contamination hinders the use of Sanger sequencing.." is not very accurate. There are two aspects here - the amplification of virus from clinical samples is one; it allows to sequence directly from the clinical samples (rather than amplification via cultures) and is addressed in this paper by the improved amplicons. The sequencing of this amplified virus is another aspect, which can be done by any sequencing technology. Here, you chose to use ONT, which is cheap, rapid and can be applied in third world countries and "in the field".
- In Discussion - "We have already sequenced 12 samples within 10 hours" is confusing since you mention in the paper that it takes 3 days (including amplification and library prep).
- In Discussion - "These Ct values demonstrated an inverse relationship with the sequencing". This needs a figure. Also it would be interesting to discuss the limit of sampling - up to which Ct value can a sample be sequenced? You mention this later in the discussion ("which a good sample quality") but it's missing a number or a description - what is considered a good sample quality?

Reviewer comments

This study presents the development of a rapid, culture-independent workflow for whole genome sequencing of Nipah virus (NiV) using Oxford Nanopore technology. It addresses two critical challenges in resource-limited settings dealing with highly infectious viruses: (1) enabling safe sequencing directly from clinical samples without viral culturing, using amplicons that span the entire genome, and (2) providing a fast and cost-effective sequencing approach with an affordable device. While the authors did not design the primers themselves, they optimized their use and demonstrate their effectiveness across relevant clinical and environmental samples with varying viral loads. Notably, this study underscores the vital role of genome sequencing in public health, particularly in resource-constrained regions facing viral outbreaks, facilitating real-time outbreak monitoring, NiV transmission investigation, and tracking viral evolution.

Comments:

- In the introduction:
“...substitutions/site/year (8) while this evolution monitoring is important...”.
change to: ...substitutions/site/year (8). **This** evolution monitoring is important
- In the introduction:
“In addition to addressing these issues, whole-genome sequencing...”
change to: To address these issues...
- In methods, 2.1 – a possible typo - “icddr,b” ?
- In methods, 2.3 – “we used two pools of primers designed by the ARTIC Network but modified by our lab experts”. How / where were the primers modified and why?
- In methods, 2.8 – what was the reasoning behind choosing only 12 samples at a time to load onto the flow cell? What’s the maximum number of positive samples that can be loaded into a single flow cell at a time?
- In methods, 2.8 - what was the reasoning for using an older-version flow-cell (R9.4.1) rather than the newer available versions (R10.4)?
- In methods, 2.9 – what’s the reasoning behind analyzing only 10,000 reads per sample?
- In results, what’s the reasoning behind running the device 12 hours (and not less/more)? Did you see that enough genomic coverage was achieved only after 12 hours? maybe time can be saved for lower Ct samples if the run stops before.
I think you should add a figure with a graph that states the coverage of the genome as a function of sequencing time and maybe even the Ct of the sample. You can refer to this new figure from parts in your results section that you write about but not represent in a figure. E.g. this sentence: “The coverage of the reference Nipah virus genome across the sequenced samples exhibited an inverse correlation with the Ct value, whereby an increase in the Ct value corresponded to a reduction in mean coverage” should be shown in a figure. I would put this figure instead of Figure 2 and put figure 2 in the supplementary.
- In results, 12 samples were run but you say that only 9 complete genomes were achieved. The phrasing of this part is very confusing and should be rephrased.
- Table 1 – please add the genomic coverage % for each sample.
- Figure 3 – belongs in the supplementary, and is not referred to in the text.

- Figure 4 is not referred to in the results section.
- In results, it would be nice to add a figure in the supplementary for this sentence: “Furthermore, the mean reading quality (Q) of these samples was lower compared to samples with lower Ct values”
- In Discussion – ONT and NGS were already defined earlier in the manuscript.
- In Discussion – you mention that one of the main advantages of ONT is that it produces long reads, however this doesn’t serve this paper, as the amplicons are 400bp in length (which is a bit longer than Illumina reads but still is not the whole genome). The main advantage of ONT in this case is that it’s cheaper and can be applied in third world countries and “in the field” if necessary.
- In Discussion – this sentence – “The limited presence of viruses in clinical samples, along with host nucleic acid contamination hinders the use of Sanger sequencing..” is not very accurate. There are two aspects here - the amplification of virus from clinical samples is one; it allows to sequence directly from the clinical samples (rather than amplification via cultures) and is addressed in this paper by the improved amplicons. The sequencing of this amplified virus is another aspect, which can be done by any sequencing technology. Here, you chose to use ONT, which is cheap, rapid and can be applied in third world countries and “in the field”.
- In Discussion – “We have already sequenced 12 samples within 10 hours” is confusing since you mention in the paper that it takes 3 days (including amplification and library prep).
- In Discussion – “These Ct values demonstrated an inverse relationship with the sequencing”. This needs a figure. Also it would be interesting to discuss the limit of sampling – up to which Ct value can a sample be sequenced? You mention this later in the discussion (“which a good sample quality”) but it’s missing a number or a description – what is considered a good sample quality?

Dear Dr Alexander Bello,

Thank you for giving us the opportunity to submit a revised version of our manuscript titled “Development of a Culture Independent Whole Genome Sequencing of Nipah Virus Using MinION Oxford Nanopore platform” (Manuscript number: **Spectrum02492-24**) to the Microbiology Spectrum. We appreciate the time and effort that you and the reviewers have dedicated to providing valuable feedback on our manuscript. We are grateful to the reviewers for their insightful comments and constructive suggestions to improve the manuscript substantially. We have incorporated changes to reflect the suggestions provided by the reviewers in the revised manuscript. We have made changes within the manuscript (without figure) using Microsoft Word’s “track changes (all marked-up)” feature.

Here is a point-by-point response to the reviewers’ comments and concerns.

Responses to the comments of Reviewer 2

Reviewer #2 (Comments for the Author):

In this article by Rahman et al, the authors design a strategy to perform Nipah virus whole genome sequencing using the long-read sequencer, ONT. The movement towards the ONT platforms for pathogen surveillance is growing in popularity and aligns well with resource limited settings, like Bangladesh described here. Overall, the article shows the feasibility of the work however I believe the though following would significantly strengthen the impact and provide clarity:

##Minor

Reviewer Point #1: In Methods 2.3 it states: "we used two pools of primers designed by the ARTIC Network but modified by our lab experts" What was modified on these primers? How do they differ from the standard ARTIC Network sets?

Author Response #1: We sincerely thank the reviewer for the insightful comments and critical suggestions to improve the manuscript. The standard ARTIC Network primer (<https://github.com/artic-network/primer-schemes>) set was originally designed for the Nipah virus Malaysia genotype (NiV-MY) using the reference sequence AJ564622. However, this primer set was not fully compatible with the Nipah virus Bangladesh genotype (NiV-BD) due to genetic differences between the two strains. To address this, we modified some nucleotide bases in the primer set to ensure compatibility with the Bangladeshi Nipah virus strain. The list of primers and their sequences are provided in **Supplementary Table 1**, where modifications in the primer sequences are highlighted in **red**.

Reviewer Point #2: The third paragraph of the results states the average read length was 554bp. It would be helpful to know the expected average for the ARTIC primers to see how well these aligned between expected and acquired reads.

Author Response #2: We thank the reviewer for this important observation. The ARTIC primer sets used in this study were designed to generate 400 bp amplicons; however, due to overlapping target sites, some amplicons exceeded this intended length. As a result, the average read length increased to 554 bp compared to the expected amplicon size (400 bp) ensures greater overlap between adjacent amplicons, facilitating accurate genome assembly and reducing the risk of dropout regions. This extension in read length may have influenced sequencing efficiency and coverage distribution, which are important considerations for optimizing future protocols. We appreciate this perspective and will further evaluate its implications in our analysis. As per the reviewer's suggestion, we have revised the Methodology and Results sections of the revised manuscript on **line 156-157 page 6 and Line 272-273, page 10** respectively.

Reviewer Point #3: Figure 3: There is no indication which samples are corresponding to each panel. Please change the titles to the sample names rather than using the same title over and over. Also the figure text (e.g. title) appear to be different sizes, please correct to keep all the same size

Author Response #3: We thank the reviewer for the valuable comments and the critical suggestions to enhance the quality of the manuscript. We have replaced the titles by sample name in each panel with the corresponding barcode as suggested by the reviewer. The samples name is now consistent with the **text and the Table 1** in the revised manuscript. We have adjusted the figure text (title) to the same size accordingly. As per the reviewer's (3rd reviewer) recommendation, we have included the **Figure 3** in the supplementary section as **Supplementary Figure 1**.

Reviewer Point #4: Table 1: This is a bit confusing. A few columns are labeled as x1000. I am unsure it would be possible to get this sequencing depth or reads for this flow cell and the number of samples pooled (e.g. it reads that sample 1 got 728,000x coverage).

Author Response #4: We thank the reviewer for pointing out the confusion in **Table 1**. It was a typographical error labeling the column of Mean Genomic Coverage with x1000. To address this, we have made changes the notation of "**Mean Genomic Coverage**" instead of "**Mean Genomic Coverage (x1000)**" in the revised **Table 1** in the revised manuscript.

Reviewer Point #5: Table 1: Rather than showing "Highest Nucleotide Identity" (or perhaps in addition to this column) I recommend you show genome coverage. This gives a false impression that sample 7, which only got 10 reads, still had >99% genome coverage.

Author Response #5: We sincerely thank the reviewer for the insightful comment and constructive suggestions to improve the manuscript. As per the reviewer's recommendation, we have added a new column titled "**Genome Coverage (%)**" along with "**Highest Nucleotide Identity**" in the revised **Table 1** in the revised manuscript.

Reviewer Point #6: Did the NTC sample have any detection of Nipah virus reads (suggesting cross-contamination). Please put the NTC data/stats in Table 1

Author Response #6: We thank the reviewer for raising a valid point to add. In our run, we did not get any NiV-specific mapped reads in the NTC (No Template Control) that underscore the reliability of the workflow and the integrity of the data generated for the study. We used nuclease-free water as an NTC for the sequencing run. As per the reviewer's suggestion, we have incorporated the data regarding the NTC sample in the revised **Table 1 as well as in the text on Lines 273-275, Page 10** in the revised manuscript.

Reviewer Point #7: ARTIC sequencing on ONT platforms isn't completely unique to this study. How do your results compare to other ARTIC protocols for other viral pathogens? Could expand in the discussion section

Author Response #7: We thank the reviewer for the constructive comments and critical suggestions to improve the manuscript. The ARTIC Network has established a robust framework for sequencing viral genomes, with protocols successfully applied to pathogens such as SARS-CoV-2, Influenza, Ebola virus, Dengue, Chikungunya and Zika virus. We humbly state that the application of ARTIC protocol to NiV is relatively novel because as of our knowledge, we are the first to sequence the raw NiV genome sequence from clinical and environmental samples without culture the virus. Our study builds upon this established approach by modifying the ARTIC primers to optimize Nipah virus whole genome sequencing. These modifications enhanced amplicon balance, minimized dropout regions, and improved sequencing efficiency, particularly for challenging samples such as bat roost urine. While ARTIC protocols have demonstrated effectiveness across multiple pathogens, pathogen-specific optimizations, such as those implemented in our study, are essential for maximizing sequencing success. **We have expanded on this aspect in the discussion section of the revised manuscript lines 407-418, Pages 14-15 as follows:** “A key modification in our approach was the optimization of ARTIC primers to enhance genome recovery for the Nipah virus (Supplementary Table 1). While ARTIC protocols have been successfully applied to other viral pathogens, such as SARS-CoV-2 and Ebola virus, the standard primer sets often require pathogen-specific adjustments to maximize coverage. Our modified primers were designed to improve amplicon balance, reduce dropout regions, and ensure consistent amplification across the Nipah virus genome. These modifications contributed to achieving 98-99% genomic coverage with a mean coverage depth ranging from 400 to 1157 for samples with Ct values up to 32, demonstrating the effectiveness of our tailored approach. This modified primer set improved sequencing efficiency and genome completeness, particularly from challenging samples such as bat roost urine. While ARTIC sequencing has been widely applied across viral genomes, pathogen-specific optimization remains crucial to overcoming challenges such as low viral load and environmental interference.”

Reviewer Point #8: ARTIC protocols make use of PCR products. How often do you predict new primer sets will need to be based on NiV mutational frequency?

Author Response #8: We thank the reviewer for raising this key point to improve the quality of the manuscript. The ARTIC protocol relies on PCR amplification to target specific regions of the viral genome, and the effectiveness of primer sets is directly influenced by the genetic variability of the target virus. The similarity matrix of the NiV indicates that the Bangladeshi NiV strains from 2001 to 2023 are

closely clustered together, showing a nucleotide identity of 98-100% with the Bangladeshi reference strain. In our study, we observed approximately 200 SNPs in the newly sequenced 11 samples compared with the Bangladesh reference strain (AY988601) sequenced in 2004, indicating a relatively low mutational rate for this virus. Given this rate, the existing primer sets are likely to remain effective for a considerable period. However, continuous monitoring of NiV's mutational frequency is essential, particularly in regions targeted by the primers. While the 200 SNPs observed are not expected to significantly affect primer performance, minor adjustments to the primer sets may be necessary as the virus evolves. Routine surveillance of NiV genomic evolution will be crucial for ensuring the long-term accuracy and reliability of the ARTIC sequencing approach.

In response to this valuable feedback, we have expanded on this aspect in the revised manuscript's 'Discussion' section, incorporating relevant references to support our assertions. These revisions can be found on **lines 419–433, Page 15, as follows:**

“Since the initial identification of Nipah virus (NiV) in Malaysia in 1999, phylogenetic analyses have delineated two major genotypes: NiV-Malaysia (NiV-MYS) and NiV-Bangladesh (NiV-BD). Comparative genomic analyses indicate that these genotypes differ by approximately 8–10% at the nucleotide level [14]. Moreover, similarity matrix analyses of NiV strains from Bangladesh collected between 2004 and 2023 demonstrate a high degree of genetic conservation, with nucleotide identities ranging from 98% to 100% relative to the Bangladeshi reference strain (AY988601) [8]. In the present study, we identified approximately 200 single nucleotide polymorphisms (SNPs) in the 11 newly sequenced samples when compared to the 2004 Bangladeshi reference strain, suggesting a relatively low mutational rate for NiV. Given this observed rate, the current primer sets are expected to remain effective for an extended period. However, ongoing genomic surveillance is imperative, particularly for monitoring mutations in regions targeted by these primers. Furthermore, our findings underscore the robustness and adaptability of the ARTIC sequencing approach, providing a valuable framework for its application in the genomic surveillance of emerging and re-emerging viral pathogens, particularly those necessitating high sensitivity for field and clinical samples.”

References:

8. *Rahman, M.Z., et al., Genetic diversity of Nipah virus in Bangladesh. International Journal of Infectious Diseases, 2021. 102: p. 144-151.*
14. *Lo, M.K., et al., Characterization of Nipah virus from outbreaks in Bangladesh, 2008–2010. Emerging infectious diseases, 2012. 18(2): p. 248.*

Reviewer Point #9: It would be really interesting to see what the minimum number of reads are required for a sufficient genome coverage. This is especially useful for ONT platforms where you can stop, flush the flow cell, and sequence again. You could do this by randomly sampling reads from the samples with

high genome coverage and increase the number of sampled reads until you reach a genome coverage threshold (e.g. 98%). Then you can calculate the average number of reads needed.

Author Response #9: We thank the reviewer for this thoughtful suggestion. We agree with the reviewer that determining the minimum number of reads required for sufficient genome coverage is an interesting and potentially valuable experiment, especially for optimizing sequencing runs on the Oxford Nanopore Technologies (ONT) platform. We humbly state that random subsampling of reads from high-coverage samples showed 10,000 reads that are sufficient to achieve $\geq 98\%$ genome coverage for high-quality Nipah virus samples. However, as Nipah virus is a Risk Group 4 (RG4) pathogen, working with live virus presents significant biosafety and logistical challenges. Given these constraints, our primary objective in this study was to establish an efficient, culture-free whole-genome sequencing approach using ONT technology to facilitate the identification of new Nipah virus strains directly from clinical and environmental samples. While the ability to pause and resume sequencing on ONT platforms provides flexibility, we decided not to pursue this specific analysis at this time. We do recognize the potential benefits of such an experiment and will certainly consider it in future studies to further refine sequencing strategies. We greatly appreciate this insightful feedback and will keep it in mind for future optimization efforts, particularly as we continue to enhance genomic surveillance and strain characterization of Nipah virus.

Reviewer Point #10: These appear to be archived samples. Were this originally sequenced with other platforms (i.e. Illumina)? If so, it would be interesting to compare the results between ONT and Illumina results.

Author Response #10: We thank the reviewer for this valuable observation. The archived samples analyzed in this study were not previously sequenced using Illumina or other platforms. Our primary objective was to optimize a culture-free sequencing method for Nipah virus whole genome sequencing using ONT, particularly for challenging sample types like bat roost urine. This work aligns with broader efforts in pandemic preparedness by enhancing genomic surveillance capabilities for high-risk pathogens like Nipah virus, which has the potential for spillover and outbreak emergence. Nonetheless, a direct comparison between ONT and Illumina sequencing would be valuable for assessing accuracy, error profiles, and coverage differences. Future studies could incorporate Illumina sequencing to validate our findings and further refine the ARTIC approach for Nipah virus genomic surveillance. We appreciate this suggestion and will consider it for potential follow-up studies, particularly in the context of strengthening early detection and response strategies for emerging viral threats..

##Major

Reviewer Point #11: In the discussion it states, "We implemented an extraction protocol involving DNase I enzyme treatment to selectively degrade host and bacterial DNA during the RNA extraction process for the clinical samples." There is no data presented regarding this approach and I did not see description of this in the methods section. I would strongly encourage that this data is included to show the efficiency of your extraction approach

Author Response #11: We thank the reviewer for raising suitable recommendations for improving the manuscript. For the RNA extraction from environmental samples, we utilized the Direct-zol RNA Miniprep Kit. We specifically chose this kit because it includes native DNase I treatment as part of the manufacturer's protocol, which effectively removes contaminating genomic DNA during the RNA

extraction process. We have revised the discussion section of the manuscript to clarify this aspect and provide a more accurate description of our approach in the revised manuscript **on page 12, lines 358-361** as follows: “We implemented an RNA extraction protocol that includes DNase I enzyme treatment, as part of the manufacturer's protocol for the Direct-zol RNA Miniprep Kit, to selectively degrade host and bacterial DNA during the extraction process for environmental samples.”

Additionally, we have previously demonstrated the effectiveness of DNase-containing kits for viral RNA extraction in our study on culture free sequencing of Influenza A virus using the ONT platform directly from clinical specimen (<https://journals.asm.org/doi/10.1128/spectrum.04946-22>). Our experience with this approach further supports its suitability for Nipah virus sequencing, particularly when working with different sample types.

Reviewer Point #12: Generally speaking, the majority of the data is showing QC information for the run metrics, which is helpful to have. However, where this publication lacks is the strength of how the data generated is useful from a pathogen genomics perspective. For example the following would substantially increase the impact:

Author Response #12: We apologize that our sequencing data is more elaborative of showing QC information for the run metrics than the data generation procedure. We have addressed the following concerns raised by the reviewer in the revised manuscript step by step.

o **Reviewer Point #12.1:** How divergent are these viruses? How many SNPs/indels were detected?

Author Response #12.1: We thank the reviewer for raising this point. Phylogenetic analysis reveals that NiV exhibits two major genotypes: NiV-Malaysia (NiV-MYS) and NiV-Bangladesh (NiV-BD). Distance matrices indicate that these genotypes differ by 8–10% at the nucleotide level and 2–8% at the amino acid level. The similarity matrix of NiV indicates that the Bangladeshi NiV strains from 2001 to 2023 are closely clustered together, showing a nucleotide identity of 98–100% relative to a Bangladeshi reference (AY988601) strain. As per the reviewer's recommendation, we have determined the SNPs and indels in the 12 sequences. We have also determined the frequency of SNPs in those genomes. We have revised the “Results” section of the revised manuscript **on page 11, lines 308-315**, as follows “Single nucleotide polymorphisms (SNPs) were identified in 11 out of 12 Nipah virus (NiV) genomes by comparing them to the Bangladeshi NiV reference sequence (Accession No: AY988601.1). Among the nine NiV strains with genome coverage exceeding 98%, the number of SNPs ranged from 164 to 206, with the frequency of around 1% (Figure 6C). The strain NINS-00123 (U) exhibited the highest SNP count (206), while RS0652823 had the lowest (164) (Figure 6B, Figure 6C). In contrast, due to lower genome coverage, RP0649923 and J106 contained fewer SNPs, with counts of 116 and 122, respectively. Additionally, no insertion or deletion mutations were identified in the 12 newly sequenced NiV genomes.” We have also incorporated **two new figures (Figure 6B and Figure 6C)** to describe the genome-wide number and frequency of the SNPs in the revised manuscript.

o **Reviewer Point #12.2:** Phylogenetic trees would be helpful here to show the diversity of viral sequences captures

Author Response #12.2: We thank the reviewer for this valuable suggestion. We have included the phylogenetic tree to visually depict the diversity of viral sequences in the revised manuscript in the “Methods and Materials” section of the revised manuscript under sub-section 2.10 titled “Phylogenetic analysis” and added appropriate references on **pages 08-09, lines 232-241** as follows:

“2.10. Phylogenetic analysis:

“A phylogenetic tree was constructed using all successfully sequenced Nipah virus genomes. Additionally, 74 whole-genome sequences (~18,200 bp) of Nipah virus strains, sequenced between 1999 and 2023 and curated from the NCBI GenBank database, were included in the analysis. Multiple sequence alignment was performed using BioEdit (version 7.2.5.0) to ensure an accurate comparison of selected sequences. The phylogenetic tree was generated using the Maximum Likelihood method in MEGA (version 11.0.13), applying the T93+G+I substitution model [21], as determined by ModelFinder [22]. The robustness of the tree was assessed through phylogenetic bootstrapping with 1,000 replicates. SNPs between the newly sequenced whole genome of NiV strains and the Bangladeshi reference genome (AY988601) were using the python-based tool, SNP-site [23]”.

In the Results section of the revised manuscript, we have incorporated these revisions “Phylogenetic analysis using the Maximum Likelihood method on 85 whole-genome NiV sequences demonstrated that Bangladeshi strains form a phylogenetically distinct lineage, further dividing into two sub-lineages. Of the 12 newly sequenced genomes, 11 were included in this analysis, all clustering within the NiV-BD2 sub-lineage (Figure 6A).” on **Page 11, Lines 304–307**, to depict the phylogenetic analysis. This analysis includes 11 out of 12 whole genome sequences from our study alongside 74 previously sequenced Nipah virus genomes obtained from the NCBI GenBank database.

References:

21. Tamura, K. and M. Nei, Estimation of the number of nucleotide substitutions in the control region of mitochondrial DNA in humans and chimpanzees. *Molecular biology and evolution*, 1993. **10**(3): p. 512-526.
22. Kalyaanamoorthy, S., et al., ModelFinder: fast model selection for accurate phylogenetic estimates. *Nature methods*, 2017. **14**(6): p. 587-589.
23. Page, A.J., et al., SNP-sites: rapid efficient extraction of SNPs from multi-FASTA alignments. *Microbial genomics*, 2016. **2**(4): p. e000056.

o **Reviewer Point #12.3:** Plotting the data showing genome coverage across the samples would be incredibly helpful. There likely are biased areas of the genome with greater depth, but there is no data showing this.

Author Response #12.3: We thank the reviewer for this valuable suggestion. As per the reviewer's recommendation, we have now added a **figure (Figure 2)** that shows genome coverage across all sequenced samples. This new figure and discussion may provide a clear understanding of genome coverage across the viral genome.

Responses to the comments of Reviewer 3

Reviewer #3 (Comments for the Author):

This study presents the development of a rapid, culture-independent workflow for whole genome sequencing of Nipah virus (NiV) using Oxford Nanopore technology. It addresses two critical challenges in resource-limited settings dealing with highly infectious viruses: (1) enabling safe sequencing directly from clinical samples without viral culturing, using amplicons that span the entire genome, and (2) providing a fast and cost-effective sequencing approach with an affordable device. While the authors did not design the primers themselves, they optimized their use and demonstrate their effectiveness across relevant clinical and environmental samples with varying viral loads. Notably, this study underscores the vital role of genome sequencing in public health, particularly in resource-constrained regions facing viral outbreaks, facilitating real-time outbreak monitoring, NiV transmission investigation, and tracking viral evolution.

Authors Response (General): We thank the reviewer for the kind feedback, constructive criticisms and useful suggestions. We have addressed all the queries/suggestions and revised the manuscript as per the reviewer's comments, this has helped improve the quality of our revised manuscript significantly.

Comments:

-Reviewer Point #1: In the introduction: "...substitutions/site/year (8) while this evolution monitoring is important...". change to: ...substitutions/site/year (8). This evolution monitoring is importan

Author Response #1: We thank the reviewer for this observation. As per the reviewer's suggestion, we have included the following corrections "This evolution monitoring is important for vaccine design and drug development." in the revised manuscript on **page 3, line 73**.

-Reviewer Point #2: In the introduction: "In addition to addressing these issues, whole-genome sequencing..." change to: To address these issues...

Author Response #2: We thank the reviewer for the valuable suggestions to improve the manuscript. As per the reviewer's suggestion, we have included the following corrections "To address these issues, whole-genome sequencing of NiV can provide information on the identity and evolution of genetic variations and the origin of this virus" in the revised manuscript on **page 3, lines 76**.

- Reviewer Point #3: In methods, 2.1 - a possible typo - "icddr,b" ?

Author Response #3: We humbly state that icddr,b is not a typo. icddr,b stands for International Centre for Diarrhoeal Disease Research, Bangladesh. We have provided the full abbreviation in the revised manuscript at first use to avoid the confusion raised by the reviewer. We have included the following corrections "International Centre for Diarrheal Disease Research, Bangladesh (icddr,b)." in the revised manuscript on **page 5, line 128**.

- Reviewer Point #4: In methods, 2.3 - "we used two pools of primers designed by the ARTIC Network but modified by our lab experts". How / where were the primers modified and why?

Author Response #4: We thank the reviewer for this excellent observation, other reviewers have also raised the points to enrich the discussion section. In this study, we have fine-tuned the default ARTIC primer sets. The standard ARTIC Network primer set (<https://github.com/artic-network/primer-schemes>) was originally designed for the Nipah virus Malaysia genotype (NiV-MYS) using the reference sequence AJ564622. However, this primer set was not fully compatible with the Nipah virus Bangladesh genotype (NiV-BD) due to genetic differences between the two strains (NiV-B and NiV-M). To address this, we modified some nucleotide bases in the primer set to ensure compatibility with the Bangladeshi Nipah virus strain (NiV-BD). The modified primers enhance amplicon balance, minimize dropout regions, and improve sequencing efficiency. The list of primers and their sequences are provided in **Supplementary Table 1**, where modifications in the primer sequences are highlighted in **red**.

- Reviewer Point #5: In methods, 2.8 - what was the reasoning behind choosing only 12 samples at a time to load onto the flow cell? What's the maximum number of positive samples that can be loaded into a single flow cell at a time?

Author Response #5: We appreciate the reviewer for this insightful comment. At the time of the study, the laboratory archive contained 12 Nipah virus-positive samples confirmed by real-time PCR. Among these, 11 were clinical samples collected from human cases, while one was a bat roost urine sample. Moreover, as mentioned in the Discussion section, on **lines 375-377, page 13** the sequencing protocol described in the manuscript is capable of processing up to 48 samples simultaneously.

- Reviewer Point #6: In methods, 2.8 - what was the reasoning for using an older-version flow-cell (R9.4.1) rather than the newer available versions (R10.4)?

Author Response #6: We thank the reviewer for raising a valid point. We humbly state that at the time of this study, the R9.4.1 flow cell was the only version available for Oxford Nanopore sequencing in our laboratory. Our protocol success with the R9.4.1 flow cell demonstrates its utility for NiV sequencing in resource-limited settings, but we strongly advocate for adopting the R10.4 flow cell in future work to maximize data quality and efficiency by reducing error rate as the new flow cell increases the accuracy in raw read (~97–99%).

- **Reviewer Point #7:** In methods, 2.9 - what's the reasoning behind analyzing only 10,000 reads per sample?

Author Response #7: We thank the reviewer for raising this question. In our study, we analyzed all the mapped reads for the 12 samples. This methodology of Nipah virus whole genome sequencing utilized all remaining sequencing reads, following quality control (QC) steps, for the construction of the Nipah virus genome. The "Methods and Materials" section of the revised manuscript has been revised as follows "For each sample, all the mapped reads were analyzed" on **line 229, Page 08**.

- **Reviewer Point #8:** In results, what's the reasoning behind running the device 12 hours (and not less/more)? Did you see that enough genomic coverage was achieved only after 12 hours? maybe time can be saved for lower Ct samples if the run stops before. I think you should add a figure with a graph that states the coverage of the genome as a function of sequencing time and maybe even the Ct of the sample. You can refer to this new figure from parts in your results section that you write about but not represent in a figure. E.g. this sentence: "The coverage of the reference Nipah virus genome across the sequenced samples exhibited an inverse correlation with the Ct value, whereby an increase in the Ct value corresponded to a reduction in mean coverage" should be shown in a figure. I would put this figure instead of Figure 2 and put figure 2 in the supplementary.

Author Response #8: We thank the reviewer for the insightful suggestions. The sequencing runtime depends on genome length, amplicon size, and viral nucleic acid concentration (viral load). We agree with the reviewer that runtime can be saved for samples with lower Ct values (higher viral loads). In this study, we used 12 samples with a wide range of Ct values (25.52 to 37.72). Therefore, we set the ONT sequencing runtime to 12 hours for NiV RNA, to get sufficient good-quality data for each sample, balancing sufficient genome coverage with minimizing pore exhaustion. This decision was made based on our prior experience with the genome sequencing of the larger SARS-CoV-2 (30 kb), where a 14-hour runtime produced high-quality data. Consequently, we initially set 12 hours of runtime for NiV as sufficient for a whole genome (~18.2 kb) sequence with $\geq 98\%$ coverage and good-quality data. We acknowledge that further optimization of sequencing runtime would be valuable; however, due to the lack of a Nipah virus culture facility, we do not have access to a sufficient quantity of samples for additional experiments. Moreover, the samples we are working with are clinical and environmental, which are often limited in quantity and difficult to obtain. The rarity of human Nipah virus infections, combined with the rapid progression and high fatality rate of the disease, further restricts opportunities for resampling, making additional experiments on run-time optimization challenging.

Nonetheless, the runtime may be reduced once sufficient data has been generated, as determined in real-time using RAMPART (Read Assignment, Mapping, and Phylogenetic Analysis in Real Time)

(<https://artic.network/rampart>). Future studies may explore further optimization of runtime as more samples become available.

Secondly, as per the reviewer's suggestion, we have introduced two scatter plots illustrating the correlation between the Ct value and the genomic coverage% (**Figure 5A**) and mean genomic coverage (**Figure 5B**). These figures visually represent the relationship between viral load (as indicated by Ct value) and sequencing depth, providing a clearer understanding of how sample quality affects coverage. Moreover, **Figure 2** has been shifted to the Supplementary section as "**Supplementary Figure 1**" from the main text in the revised manuscript.

- **Reviewer Point #9:** In results, 12 samples were run but you say that only 9 complete genomes were achieved. The phrasing of this part is very confusing and should be rephrased.

Author Response #9: We thank the reviewer for this valuable observation. As per the reviewer's suggestion, we have re-written sentences as "A 12-hour sequencing run was performed using the pooled DNA library on the FLO-MIN106D (R9.4.1) flow cell of the Oxford Nanopore MinION MK 1C platform. This sequencing approach successfully generated complete genomes for nine out of twelve Nipah virus-positive samples, with coverage ranging from 98.01% (Ct 29.26 ± 8.54) to 99.79% (Ct 22.34). Among the remaining three samples, two with high Ct values ($30 \leq Ct \leq 35$) yielded partial genome coverage between 70.02% and 86.05%. However, for samples with $Ct \geq 35$, near-complete genome coverage could not be achieved (Table 1, Figure 2)" in the revised manuscript on lines 258-265 pages 09.

- **Reviewer Point #10:** Table 1 - please add the genomic coverage % for each sample.

Author Response #10: We thank the reviewer for this insightful suggestion. We have added the genomic coverage percentage for each sample in **Table 1** in the revised manuscript on **line 547, Page 18**.

- **Reviewer Point #11:** Figure 3 - belongs in the supplementary, and is not referred to in the text.

Author Response #11: We thank the reviewer for this great observation. We have shifted **Figure 3** to the supplementary section (**Supplementary Figure 1**) and have referred to it in the text in the revised manuscript as suggested on **line 272, Page 10**.

- **Reviewer Point #12:** Figure 4 is not referred to in the results section.

Author Response #12: We sincerely thank the reviewer for this observation. We have referred to **Figure 4** in the revised text in the Results section of the revised manuscript on **page 10 and lines 271** as suggested by the reviewer.

- **Reviewer Point #13:** In results, it would be nice to add a figure in the supplementary for this sentence:

"Furthermore, the mean reading quality (Q) of these samples was lower compared to samples with lower Ct values"

Author Response #13: We thank the reviewer for this valuable suggestion. As per the reviewer's recommendation, we have included a scatter plot of Ct values versus mean read quality scores (Q) (**Supplementary Figure 2**) and referred this figure in the revised manuscript on **line 270, page 10**.

- **Reviewer Point #14:** In Discussion - ONT and NGS were already defined earlier in the manuscript.

Author Response #14: We thank the reviewer for this observation. We agree with the reviewer that ONT and NGS were defined earlier, so we have excluded the ONT and NGS from the discussion section to avoid redundancy and enhance the manuscript's clarity as suggested by the reviewer in the revised manuscript on **lines 330 & 332, Page 12**.

- **Reviewer Point #15:** In Discussion - you mention that one of the main advantages of ONT is that it produces long reads, however this doesn't serve this paper, as the amplicons are 400bp in length (which is a bit longer than Illumina reads but still is not the whole genome). The main advantage of ONT in this case is that it's cheaper and can be applied in third world countries and "in the field" if necessary.

Author Response #15: We thank the reviewer for raising suitable recommendations for improving the manuscript. We agree with the reviewer that the main advantages of ONT are its cost-effectiveness and its applicability for use in field settings, not the long-read lengths, as the amplicons are approximately 400 bp in length. While ONT's long reads are beneficial for many applications, particularly when full-length viral genomes or complex genomes are being sequenced, in our case, the key advantages of using ONT are its lower cost and its potential for application in resource-limited settings, including fieldwork in low- and middle-income countries (LMICs). The affordability and portability of ONT technology make it an attractive option for genomic surveillance, especially for emerging and re-emerging infectious diseases where timely and widespread testing is critical. These factors, combined with the ease of use and the relatively low technical infrastructure required, make ONT sequencing a valuable tool for on-site detection and monitoring of viruses such as the Nipah virus. As recommended by the reviewer, we have updated the discussion section as "The major advantages of ONT sequencing in this study is its affordability and the potential for application in resource-limited settings, including field conditions. While ONT is often recognized for generating long reads, the amplicons used in our study are approximately 400 bp in length, and do not encompass the entire viral genome. As such, the main benefit of ONT in this context lies in its cost-effectiveness" to include these points in the revised manuscript on **lines 434-438, Page 15**.

- **Reviewer Point #16:** In Discussion - this sentence - "The limited presence of viruses in clinical samples, along with host nucleic acid contamination hinders the use of Sanger sequencing." is not very accurate. There are two aspects here - the amplification of virus from clinical samples is one; it allows to sequence directly from the clinical samples (rather than amplification via cultures) and is addressed in this paper by the improved amplicons. The sequencing of this amplified virus is another aspect, which can be done by any sequencing technology. Here, you chose to use ONT, which is cheap, rapid and can be applied in third world countries and "in the field".

Author Response #16: We thank the reviewer for this observation. Based on the reviewer's feedback, we have omitted the sentence (**line 343-345, page 12**) to resolve the misunderstanding of our assertion and further discussed the cost-effectivity of the ONT platform as well as its ease of use in the field as per the reviewer suggestion as follows: "The major advantages of ONT sequencing in this study is its affordability and the potential for application in resource-limited settings, including field conditions. While ONT is often recognized for generating long reads, the amplicons used in our study are approximately 400 bp in length and do not encompass the entire viral genome. As such, the main benefit of ONT in this context lies in its cost-effectiveness." in the revised manuscript on **lines 434-438, Page 15**.

- **Reviewer Point #17:** In Discussion - "We have already sequenced 12 samples within 10 hours" is confusing since you mention in the paper that it takes 3 days (including amplification and library prep).

Author Response #17: We thank the reviewer for pointing out the confusion. For 12 samples, the entire process, from nucleic acid extraction to sequence data acquisition, takes almost 3 days, whereas the nanopore sequencing run takes around 12 hours. Moreover, our workflow is scalable; we can process 48 samples at a time, allowing us to load up to 48 samples for sequencing in a single run, which can also take 3 days. As per the reviewer's concerns, the following information has been added to the discussion section "We have sequenced 12 samples with 12 hours of sequencing run, and this protocol can accommodate up to 48 samples at a time." and "From nucleic acid extraction to sequence data acquisition, takes almost 3 days, whereas the nanopore sequencing run takes around 12 hours" of revised manuscript on **Page 13, Line 375-377 and page 14 Line 383-384** respectively.

- **Reviewer Point #18:** In Discussion - "These Ct values demonstrated an inverse relationship with the sequencing". This needs a figure. Also it would be interesting to discuss the limit of sampling - up to which Ct value can a sample be sequenced? You mention this later in the discussion ("which a good sample quality") but it's missing a number or a description - what is considered a good sample quality?

Author Response #18: We thank the reviewer for pointing out the important issue. As per the reviewer's recommendation, we have included **two figures (Figure 5A and Figure 5B)** in the revised manuscript illustrating the inverse relationship between Ct value and genomic coverage in the revised manuscript. Our study found that samples with lower Ct values yielded higher genomic coverage. From our data, we also recommend that a Ct value less than 30 is considered a good sample for sequencing, providing high-quality genome coverage of over 100×. However, the limited number of positive NiV samples and the absence of a virus culture facility prevented us from establishing a definitive correlation between Ct value and sequencing quality, as well as determining the sampling limit.

Re: Spectrum02492-24R1 (Development of a Culture Independent Whole Genome Sequencing of Nipah Virus Using MinION Oxford Nanopore platform)

Dear Dr. Mohammed Ziaur Ziaur Rahman:

Your manuscript has been accepted, and I am forwarding it to the ASM production staff for publication. Your paper will first be checked to make sure all elements meet the technical requirements. ASM staff will contact you if anything needs to be revised before copyediting and production can begin. Otherwise, you will be notified when your proofs are ready to be viewed.

Sincerely,
Alexander Bello
Editor
Microbiology Spectrum

Reviewer #3 (Comments for the Author):

my comments have been addressed in full

Reviewer comments

This study presents the development of a rapid, culture-independent workflow for whole genome sequencing of Nipah virus (NiV) using Oxford Nanopore technology. It addresses two critical challenges in resource-limited settings dealing with highly infectious viruses: (1) enabling safe sequencing directly from clinical samples without viral culturing, using amplicons that span the entire genome, and (2) providing a fast and cost-effective sequencing approach with an affordable device. While the authors did not design the primers themselves, they optimized their use and demonstrate their effectiveness across relevant clinical and environmental samples with varying viral loads. Notably, this study underscores the vital role of genome sequencing in public health, particularly in resource-constrained regions facing viral outbreaks, facilitating real-time outbreak monitoring, NiV transmission investigation, and tracking viral evolution.

Comments:

- In the introduction:
“...substitutions/site/year (8) while this evolution monitoring is important...”.
change to: ...substitutions/site/year (8). **This** evolution monitoring is important
- In the introduction:
“In addition to addressing these issues, whole-genome sequencing...”
change to: To address these issues...
- In methods, 2.1 – a possible typo - “icddr,b” ?
- In methods, 2.3 – “we used two pools of primers designed by the ARTIC Network but modified by our lab experts”. How / where were the primers modified and why?
- In methods, 2.8 – what was the reasoning behind choosing only 12 samples at a time to load onto the flow cell? What’s the maximum number of positive samples that can be loaded into a single flow cell at a time?
- In methods, 2.8 - what was the reasoning for using an older-version flow-cell (R9.4.1) rather than the newer available versions (R10.4)?
- In methods, 2.9 – what’s the reasoning behind analyzing only 10,000 reads per sample?
- In results, what’s the reasoning behind running the device 12 hours (and not less/more)? Did you see that enough genomic coverage was achieved only after 12 hours? maybe time can be saved for lower Ct samples if the run stops before.
I think you should add a figure with a graph that states the coverage of the genome as a function of sequencing time and maybe even the Ct of the sample. You can refer to this new figure from parts in your results section that you write about but not represent in a figure. E.g. this sentence: “The coverage of the reference Nipah virus genome across the sequenced samples exhibited an inverse correlation with the Ct value, whereby an increase in the Ct value corresponded to a reduction in mean coverage” should be shown in a figure. I would put this figure instead of Figure 2 and put figure 2 in the supplementary.
- In results, 12 samples were run but you say that only 9 complete genomes were achieved. The phrasing of this part is very confusing and should be rephrased.
- Table 1 – please add the genomic coverage % for each sample.
- Figure 3 – belongs in the supplementary, and is not referred to in the text.

- Figure 4 is not referred to in the results section.
- In results, it would be nice to add a figure in the supplementary for this sentence: "Furthermore, the mean reading quality (Q) of these samples was lower compared to samples with lower Ct values"
- In Discussion – ONT and NGS were already defined earlier in the manuscript.
- In Discussion – you mention that one of the main advantages of ONT is that it produces long reads, however this doesn't serve this paper, as the amplicons are 400bp in length (which is a bit longer than Illumina reads but still is not the whole genome). The main advantage of ONT in this case is that it's cheaper and can be applied in third world countries and "in the field" if necessary.
- In Discussion – this sentence – "The limited presence of viruses in clinical samples, along with host nucleic acid contamination hinders the use of Sanger sequencing.." is not very accurate. There are two aspects here - the amplification of virus from clinical samples is one; it allows to sequence directly from the clinical samples (rather than amplification via cultures) and is addressed in this paper by the improved amplicons. The sequencing of this amplified virus is another aspect, which can be done by any sequencing technology. Here, you chose to use ONT, which is cheap, rapid and can be applied in third world countries and "in the field".
- In Discussion – "We have already sequenced 12 samples within 10 hours" is confusing since you mention in the paper that it takes 3 days (including amplification and library prep).
- In Discussion – "These Ct values demonstrated an inverse relationship with the sequencing". This needs a figure. Also it would be interesting to discuss the limit of sampling – up to which Ct value can a sample be sequenced? You mention this later in the discussion ("which a good sample quality") but it's missing a number or a description – what is considered a good sample quality?